# PureCover: Bridging the Gap in Re-ranking for Retrieval-Augmented Generation via Balancing Coverage and Noise

## Abstract

Re-ranking, originating from Information Retrieval (IR), has become a critical technique for filtering retrieved documents in Retrieval-Augmented Generation (RAG). Current RAG systems often directly apply re-rankers from traditional IR, which were originally designed to provide relevant and diverse documents to human users. However, this adoption overlooks a fundamental gap: unlike humans can use selective attention to filter noise and focus on key evidence, LLMs lack this ability. This gap causes traditional re-rankers to fail in covering essential evidence and minimizing noise for LLMs, significantly hurting RAG performance, especially in complex question-answering tasks. To address this, we argue that RAG re-rankers should serve a distinct objective: not only ensuring the coverage of key information but also minimizing noise in the selected document set. To achieve this objective, we propose **PureCover**, a document selection framework tailored for RAG. Instead of relying on traditional Top-$K$ re-ranking, we reformulate the document selection process as a multi-objective optimization problem and solve it by exploiting LLM attention patterns during goal-oriented reasoning. To improve efficiency, we distill the selection capability into an LLM selector via a set-wise strategy. Experiments on four multi-hop QA benchmarks demonstrate that PureCover consistently outperforms state-of-the-art baselines, achieving a better balance between coverage and noise for RAG.

## 1 Introduction

Originating in Information Retrieval (IR), re-ranking has become a core technique for filtering retrieved documents in Retrieval-Augmented Generation (RAG) (Gao et al., 2023; Zhao et al., 2023). In traditional IR systems (e.g., search engines), re-rankers aim to present relevant and diverse documents to users, allowing them to integrate information from useful documents (Carbonell & Goldstein, 1998; Wu et al., 2024). This design primarily relies on humans' selective attention mechanism, which enables them to actively focus on key information while ignoring query-irrelevant ones during information-seeking tasks (Müller & Krummenacher, 2006) (see Figure 1 (a) for an example).

With the rise of Large Language Models (LLMs), these re-rankers are increasingly used to select documents for LLMs in RAG systems rather than human users (Zhao et al., 2023; Gao et al., 2023). This shift exposes a critical gap: unlike human beings, who can focus on key information actively, LLMs do not exhibit such selective attention, making them struggle to focus on key information and more vulnerable to irrelevant documents (Shi et al., 2023; Cuconasu et al., 2024). Nevertheless, most RAG systems directly adopt traditional IR re-rankers (Qi et al., 2020; Glass et al., 2022; Li et al., 2024), whether relevance-oriented (Xiao et al., 2024; Zhang et al., 2025) or coverage-oriented (Santos et al., 2010; Carbonell & Goldstein, 1998), without addressing this gap.

Our empirical study finds that directly applying traditional IR re-rankers into RAG systems often leads to significant performance degradation, especially on complex question answering (QA) tasks (e.g., multi-hop QA). As shown in Figure 1 (b), relevance-oriented re-rankers (Xiao et al., 2024), which prioritize query-document relevance, often yield insufficient coverage of the essential information required to answer complex queries (Lee et al., 2025). Conversely, coverage-oriented re-rankers promote diversity across subtopics but inevitably introduce noisy documents that distract

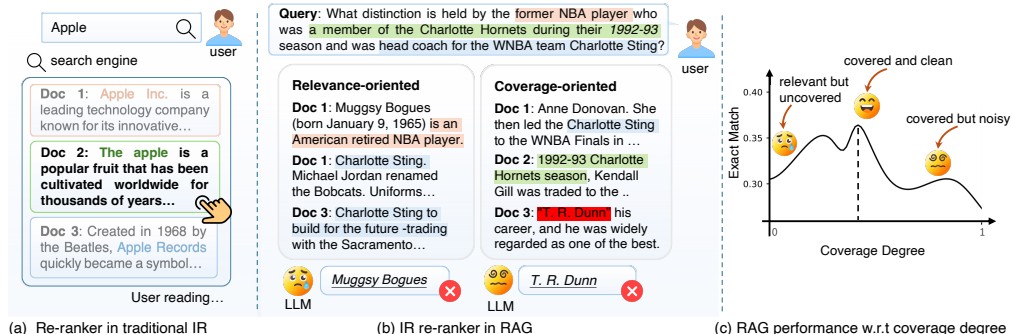

(a) Re-ranker in traditional IR  (b) IR re-ranker in RAG  (c) RAG performance w.r.t coverage degree

Figure 1: (a) Traditional IR re-rankers rely on humans' selective attention: users can naturally extract key information from a diversified ranking list. For example, a user searching for "Apple" with the intent "apple nutrition facts" can naturally ignore results about "Apple Inc." or "Apple Records" and focus on information about the fruit. (b) LLMs, however, lack this mechanism and are vulnerable to noise, making it challenging to apply traditional IR re-rankers to RAG directly. (c) This gap points to a distinct objective for RAG: unlike in IR, a re-ranker for RAG should not only ensure coverage of key information but also minimize noise in the re-ranked list.

the generator and amplify hallucinations (Chen et al., 2023) (Cossio, 2025). Experiments on Hot-potQA (Yang et al., 2018) (Figure 1(c)) confirm that both strategies failed to balance the trade-off between coverage and noise, causing significant performance drops in RAG.

Based on this insight, we argue that re-ranking in RAG should adopt a distinct objective: jointly maximizing the coverage of key evidence while minimizing irrelevant noise for LLM generators. This objective fundamentally differs from the diversity-oriented coverage in traditional IR (Wu et al., 2024), motivating a new formulation. Building on this, we recast the traditional Top-$K$ re-ranking strategy (Zehlike et al., 2017) as a multi-objective optimization problem (Deb et al., 2016) and redefine coverage and noise in the context of RAG, emphasizing the capture of essential information requirements while filtering out unhelpful documents. To address this optimization problem, we propose a novel training framework, **PureCover**, which leverages LLM attention patterns during reasoning to achieve an effective and efficient balance between coverage and noise.

Solving this multi-objective optimization problem faces two key challenges: (1) uncovering the underlying information requirements of complex queries, and (2) selecting documents that achieve broad coverage of these requirements with minimal noise. To address these challenges, we first infer the query's information requirements through Chain-of-Thought (CoT) reasoning and estimate the associated quantities from reasoning attention patterns. By recast the objective as a binary optimization problem , we propose a greedy algorithm that efficiently maximizes coverage while minimizing noise. To improve efficiency, we further introduce a set-wise distillation method that transfers the document selection capability into an LLM selector for efficient inference. We evaluate our method on four multi-hop QA benchmarks, and the experimental results demonstrate that PureCover consistently achieves significant improvement over state-of-the-art baselines.

In summary, our contributions are as follows. (1) We emphasize a fundamental gap between the re-ranking objectives in traditional IR and RAG, which arises from the selective attention mechanism of human users. (2) To address this gap, we introduce PureCover, a novel document selection framework, which formulates document selection in RAG as a multi-objective optimization problem and leverages LLM reasoning attention patterns to effectively address this problem. (3) Extensive experiments on four multi-hop QA datasets demonstrate the effectiveness of PureCover.

## 2 FORMULATION

### 2.1 RETRIEVAL-AUGMENTED GENERATION

RAG systems aim to tackle a knowledge-intensive task (e.g., multi-hop QA), where the dataset consists of multiple query–answer pairs. Each instance consists of a raw query $q$ paired with its corresponding ground-truth answer $y$. A RAG system $f(\cdot)$ takes a query $q$ as input and returns

an answer $\hat{y}_q = f(q)$. Specifically, for each query $q$, the RAG process begins with a retriever $\mathcal{R}_{\text{retrieve}}$ that first retrieves a candidate set of documents $\mathcal{D}_{\text{retrieve}} = \{d_1, d_2, \ldots, d_{K_1}\}$ with size $K_1$. To eliminate contextually irrelevant noise and cover critical information to address the query, the initially retrieved documents $\mathcal{D}_{\text{retrieve}}$ will be further filtered by the reranker $\mathcal{R}_{\text{rerank}}$ into a smaller set $\mathcal{D}_{\text{rerank}} \subseteq \mathcal{D}_{\text{retrieve}}$. Due to the generator's input limit, the re-ranker output is constrained by a budget $|\mathcal{D}_{\text{rerank}}| \leq K$. Finally, the filtered documents $\mathcal{D}_{\text{rerank}}$, along with the original query $q$, will be fed into the LLM generator as contextual information to help generate the final response $\hat{y}$.

## 2.2 RE-RANKER IN TRADITIONAL IR

**Relevance-oriented re-ranking.** The traditional relevance-oriented re-rankers, e.g., cross-encoders (Xiao et al., 2024) or LLM-based (Sun et al., 2023; Xiao et al., 2024; Zhang et al., 2025), assign an independent relevance score $\text{Rel}(q, d)$ to each document $d$ using a relevance estimation function $\text{Rel}(\cdot)$, and select the top-$K$ documents based on these scores:

$$\mathcal{D}_{\text{rerank}} = \{d_j | d_j \in \text{Top-}K\left(\text{Rel}\left(q, d\right)\right)\}. \tag{1}$$

**Coverage-oriented re-ranking.** To reduce topical redundancy and address query ambiguity, traditional IR (Carbonell & Goldstein, 1998) introduces a coverage objective $\text{Cover}'(\cdot)$ alongside relevance $\text{Rel}(\cdot)$ in re-ranking. These methods aim to select the top-$K$ documents that jointly optimize relevance and coverage in the re-ranked list, ultimately enhancing user interactions.

$$\mathcal{D}_{\text{rerank}} = \underset{\mathcal{D} \subseteq \mathcal{D}_{\text{retrieve}}, \; |\mathcal{D}| = K}{\arg\max} \Big[ \sum_{d \in \mathcal{D}} \text{Rel}(d, q) + \beta \cdot \text{Cover}'(\mathcal{D} \mid q) \Big], \tag{2}$$

where $\text{Cover}'(\mathcal{D} \mid q)$ measures how well the re-ranked document set $\mathcal{D}$ covers the underlying query intents or subtopics of $q$ (e.g., the query "Apple" may contain intents such as "Apple Inc." or "Apple Records"), and $\beta$ is a hyper-parameter that controls the trade-off between relevance and coverage.

**Challenges.** Both relevance-oriented and coverage-oriented re-ranking strategies rely on humans' selective attention mechanism (Müller & Krummenacher, 2006), which enables human users to concentrate on key evidence while disregarding noise. LLMs, however, lack this mechanism. Therefore, re-ranking in RAG requires a new objective that explicitly balances covering essential information and suppressing irrelevant content for complex QA.

## 3 OUR APPROACH: PURECOVER

To enhance RAG performance, we optimize document selection by balancing coverage and noise, framing it as a multi-objective optimization problem (Section 3.1). We then introduce PureCover, a three-step training framework (Section 3.2): (1) prompting the LLM to reason over retrieved documents and estimate key quantities from attention signals; (2) reformulating the optimization as a binary problem and solving it efficiently with a greedy algorithm; and (3) applying set-wise distillation to transfer the selection capability into a LLM-based re-ranker. The overall workflow is illustrated in Figure 2.

## 3.1 OPTIMIZATION OBJECTIVE CONSTRUCTION

In this section, we construct the objective of our re-ranker to optimize the coverage and noise in the selected document set. Formally, given the initially retrieved documents $\mathcal{D}_{\text{retrieve}}$, the selection can be expressed as the following multi-objective optimization problem:

$$\mathcal{D}_{\text{rerank}} = \underset{\mathcal{D} \subseteq \mathcal{D}_{\text{retrieve}}, \; |\mathcal{D}| \leq K}{\arg\max} \left( \text{Cover}(\mathcal{D} \mid q), \; -\text{Noise}(\mathcal{D} \mid q) \right) \tag{3}$$

where $\text{Cover}(\mathcal{D}|q)$ measures how well the selected documents collectively cover all the information requirements to answer the query $q$, and $\text{Noise}(\mathcal{D}|q)$ measures the document noise in the selected document set. Next, we formally define coverage and noise in the context of complex QA tasks.

For a complex or compositional user query $q$, there are typically multiple underlying information requirements that must be satisfied to correctly answer the query (Lee et al., 2025). We denote these requirements as $\mathcal{E}_q = \{e_1, e_2, \ldots, e_k\}$, obtained via an information need identification mechanism.

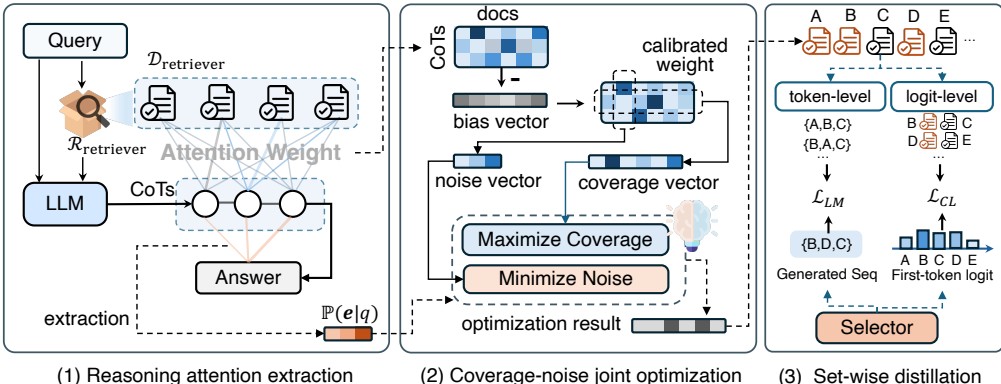

Figure 2: The overall workflow of PureCover consists of three stages: (1) We conduct a reasoning process to identify the information requirement, and extract the attention pattern; (2) We leverage the attention pattern to solve the multi-objective optimization problem; (3) To enable efficient inference, we distill the optimized result into a student model using a set-wise distillation.

Each requirement $e \in \mathcal{E}_q$ is associated with a weight $\mathbb{P}(e|q)$, such that $\sum_{e \in \mathcal{E}_q} \mathbb{P}(e|q) = 1$, representing its relative importance in addressing the query.

**Information coverage.** We define the information coverage of a selected subset $\mathcal{D}$ as the weighted expected probability that each information requirement $e$ is addressed by at least one document:

$$\text{Cover}(\mathcal{D} \mid q) = \sum_{e \in \mathcal{E}_q} \mathbb{P}(e \mid q) \cdot \mathbb{P}(E_{\mathcal{D},e}), \quad \text{where } \mathbb{P}(E_{\mathcal{D},e}) = \left[ 1 - \prod_{d \in \mathcal{D}} \left( 1 - \mathbb{P}(d \mid e) \right) \right], \quad (4)$$

where $E_{\mathcal{D},e}$ denotes the event that information requirement $e$ is satisfied by at least one document in $\mathcal{D}$, $\mathbb{P}(d|e)$ is the likelihood that document $d$ provides relevant content to satisfy the information requirement $e$, reflecting the document's utility for that specific need. This definition reflects the intuitive notion that the selected document set provides high coverage if it collectively addresses all information requirements for answering the query.

**Document noise.** Including documents that are not helpful to answer the query can substantially distract the generator and degrade the performance of RAG systems (Shi et al., 2023; Niu et al., 2025). To capture this, we define the noise of the selected set $\mathcal{D}$ as the overall disutility (i.e., unhelpfulness) within the set:

$$\text{Noise}(\mathcal{D} \mid q) = \sum_{d \in \mathcal{D}} \text{DisUtil}(d \mid q), \quad \text{where } \text{DisUtil}(d \mid q) = 1 - \mathbb{P}(d \mid e_d^*) \, \mathbb{P}(e_d^* \mid q), \quad (5)$$

where $e_d^* = \arg\max_{e \in \mathcal{E}_q} \mathbb{P}(d|e)\mathbb{P}(e|q)$ denotes the information requirement that document $d$ most significantly helps satisfy for answering the query. The disutility is high if this document poorly addresses the requirement or if the requirement itself is unlikely to be helpful to the query.

### 3.2 Objective Learning and Optimization

To solve the optimization problem in Equation 3, we leverage the attention pattern of LLMs to effectively estimate the necessary variables required in the objective. Building on this estimation, we further propose a greedy algorithm to efficiently solve the problem, and distill the results into a lightweight selector for efficient inference.

#### 3.2.1 Learning Variables via Reasoning Attention

In the optimization objective (Equation 3), two key quantities are required: the information requirement $\mathbb{P}(e|q)$ and the degree to which each document supports it $\mathbb{P}(d|e)$. While prior studies have shown that LLMs' attention patterns can highlight relevant information within long contexts (Bensemann et al., 2022; Kozlova et al., 2024; Chen et al., 2024), we extend this insight by leveraging reasoning attention patterns to explicitly estimate these two quantities. This enables us to extract the underlying information requirements of a query and evaluate document support in a way tailored

for RAG. Specifically, given a training query $q$ and its retrieved documents $\mathcal{D}_{\text{retrieve}}$, we prompt the LLM to perform goal-oriented reasoning over these documents until producing the final answer $a$: $(\mathcal{C}_q, a) = \text{LLM}[P(q, \mathcal{D}_{\text{retrieve}})]$ where $\mathcal{C}_q = \{c_1, c_2, \ldots, c_N\}$ is the reasoning sequence, with each step $c_i$ corresponding to an information requirement $e_i$. Prompt details $P$ in Appendix F.

**Attention extraction.** We leverage the token-level attention weights $\mathbf{A}$ recorded during the reasoning process as a crucial signal for downstream document selection. (See Appendix D for computational details of $\mathbf{A}$). Specifically, the token-level attention weight $\mathbf{A}_{ij}$ quantifies how much token $i$ attends to token $j$. To aggregate this information at the CoT step and document level, we define the attention from each reasoning step $c \in \mathcal{C}_q$ to each retrieved document $d_k \in \mathcal{D}_{\text{retrieve}}$ as the average token-level attention between the tokens in the step and the tokens in the document: $\text{Attn}(c, d_k) = \frac{1}{|c||d_k|} \sum_{i \in c} \sum_{j \in d_k} \mathbf{A}_{ij}$, where $|c|$ and $|d_k|$ denote the number of tokens in CoT step $c$ and document $d_k$, respectively.

**Calibration.** Raw attention weights are known to be susceptible to position biases in long contexts (Wu et al., 2025; Wan et al., 2025), such as the *lost-in-the-middle* effect (Liu et al., 2023). Using uncalibrated attention directly can therefore lead to inaccurate estimation of CoT-to-document scores. To address this, we adopt a calibration procedure following prior work (Liu et al., 2023) to remove the influence of position bias. Concretely, we define a precomputed position bias using fixed dummy CoT steps $\tilde{\mathcal{C}}$ and dummy documents $\tilde{d}_k$ and employ it to calibrate the raw attention weight assigned from CoT step $c$ to document $d_k$ located at position $k$:

$$\text{CalAttn}(c, d_k) = \text{Attn}(c, d_k) - \text{Bias}(k), \quad \text{where} \quad \text{Bias}(k) = \frac{1}{|\tilde{\mathcal{C}}|} \sum_{c \in \tilde{\mathcal{C}}} \text{Attn}(c, \tilde{d}_k), \quad (6)$$

where $\text{CalAttn}(c, d_k)$ is the position-calibrated attention from CoT step $c$ to document $d_k$. Since $\tilde{\mathcal{C}}$ and $\tilde{d}_k$ are fixed, $\text{Bias}(k)$ can be precomputed once and reused for all queries, efficiently removing position-dependent distortions.

**Estimation.** After calibration, we leverage the attention signal to estimate the two key variables (i.e., $\mathbb{P}(e_i|q)$ and $\mathbb{P}(d|e_i)$). We define the attention-based likelihood as the normalized attention weight from answer $a$ to each CoT step $c_i$ (corresponding to $e_i$) and CoT step to each document $d$:

$$\begin{cases} \mathbb{P}_{\text{attn}}(e_i \mid q) = \dfrac{\exp\left(\text{Attn}(a, c_i)/\tau_1\right)}{\sum_{m=1}^{K_1} \exp\left(\text{Attn}(a, c_i)/\tau_1\right)}, & c_i \in \mathcal{C}_q, \\[3mm] \mathbb{P}_{\text{attn}}(d \mid e_i) = \dfrac{\exp\left(\text{CalAttn}(c, d)/\tau_2\right)}{\sum_{d' \in \mathcal{D}_{\text{retrieve}}} \exp\left(\text{CalAttn}(c, d')/\tau_2\right)}, & d \in \mathcal{D}_{\text{retrieve}}, \ c_i \in \mathcal{C}_q, \end{cases} \quad (7)$$

where $\tau_1$ and $\tau_2$ is the temperature coefficient that controls the sharpness of the distribution by emphasizing documents and information requirements with higher attention weights.

### 3.2.2 COVERAGE-NOISE JOINT OPTIMIZATION

After estimating the unknown variables in the objective function, we aim to efficiently solve the optimization problem in Equation 3. To achieve this, we formulate the objective as a 0-1 integer optimization problem in Theorem 1.

**Theorem 1** (**Document selection as 0-1 integer optimization**). *By defining the coverage function and noise function in Equations 4 and 5, the denoised coverage-aware document task in Equation 3 can be formulated as a 0-1 integer (binary) multi-objective optimization problem:*

$$\max_{\mathbf{x}} \underbrace{\sum_{i=1}^{|\mathcal{E}|} \mathbf{e}_i \left[ 1 - \prod_{d=1}^{K_1} \left(1 - \mathbf{W}_{d,i}\mathbf{x}_d\right) \right]}_{Coverage} - \lambda \underbrace{\sum_{j=1}^{K_1} \left(1 - \max(\mathbf{W}_j \odot \mathbf{e})\right) \mathbf{x}_j}_{Noise} \quad (8)$$

$$s.t. \quad \sum_i \mathbf{x}_i \le K, \quad \mathbf{x}_i \in \{0, 1\} \quad \forall i \in [1, 2, \cdots, K_1],$$

*where $\mathbf{x}$ is the binary decision vector that determines each retrieved document $d_i \in \mathcal{D}_{rerank}$ is selected or not, i.e., $\mathbf{x}_i = 1$ if it is added to the re-ranking set $\mathcal{D}_{rerank}$, otherwise $\mathbf{x}_i = 0$. The constraint*

*requires that the number of selected documents does not exceed the budget $K$. $\mathbf{e}_i = \mathbb{P}_{attn}(e_i|q)$ and $\mathbf{W}$ denotes the document–requirement adjacency matrix, with entries $\mathbf{W}_{i,j} = \mathbb{P}_{attn}(d_i|e_j)$. $\max(\cdot)$ returns the largest entry in the input vector.*

**Proposition 1.** *The objective function defined in Equation 8 is monotone and submodular. Therefore, the corresponding maximization problem is NP-hard.*

The proofs of Theorem 1 and Proposition 1 are provided in Appendix E. Based on Proposition 1, this optimization problem is NP-hard. Motivated by the classical result of Nemhauser et al. (1978), which guarantees that a greedy algorithm achieves a $(1-1/e)$ approximation for monotone submodular maximization under a cardinality constraint, we design a greedy re-ranking algorithm tailored to our coverage-noise objective. At each iteration, the algorithm selects the document that yields the largest marginal gain in the objective function (Equation 8) and adds it to the re-ranked set, as detailed in Algorithm 1.

### 3.2.3 SET-WISE DISTILLATION

Direct LLM reasoning is often computationally prohibitive and slow for real-world re-ranking applications. To overcome this efficiency barrier, we introduce a set-wise distillation, which transfers the complex, optimized document selection objective into a more efficient LLM-based selector for fast inference. For a query $q$ and its retrieved documents $\mathcal{D}_{\text{retrieve}}$, we leverage teacher attention (via Equation 8 and Algorithm 1) to obtain an optimized subset $\mathcal{D}_{\text{rerank}} = \{d_1, \ldots, d_m\}$. Let $t_i$ denote the identifier of $d_i$ and $s_i$ the first-token logit from the student model. The selector (student model) is trained using a composite loss that captures this set-valued selection:

• *Permutation-invariant language modeling loss* $\mathcal{L}_{\text{LM}}$: Since the target is a set, the loss is minimized over all possible valid sequences $\mathcal{Y}$, which contains all permutations of the target document identifier sequence $y$: $\mathcal{L}_{\text{LM}} = -\min_{y' \in \mathcal{Y}} \log p_\theta(y' \mid q, \mathcal{D}_{\text{retrieve}})$

• *Contractive learning loss* $\mathcal{L}_{\text{CL}}$: This loss enforces a clear distinction between the selected documents ($d_i \in \mathcal{D}_{\text{rerank}}$) and the rejected ones ($d_j \in \mathcal{D}_{\text{retrieve}} \setminus \mathcal{D}_{\text{rerank}}$) by maximizing the score difference (logit difference $s_i - s_i$) for positive pairs $\mathcal{P}$: $\mathcal{L}_{\text{CL}} = -\frac{1}{|\mathcal{P}|} \sum_{(i,j) \in \mathcal{P}} \log \sigma(s_i - s_j)$, where $\mathcal{P}$ is the set of positive and negative document pairs: $\mathcal{P} = \{(i,j)|d_i \in \mathcal{D}_{\text{rerank}}, d_j \in \mathcal{D}_{\text{retrieve}} \setminus \mathcal{D}_{\text{rerank}}\}$. The document selector is trained with the final objective $\mathcal{L}_{\text{Final}} = \mathcal{L}_{\text{CL}} + \lambda \mathcal{L}_{\text{LM}}$.

**Inference.** At inference, candidate documents in $\mathcal{D}_{\text{retrieve}}$ are efficiently selected based on the first-token logits $s_j$, retaining only those with high logit value, i.e., $\mathcal{D}_{\text{rerank}} = \{d_j \in \mathcal{D}_{\text{retrieve}} \mid s_j > \tau_3\}$. This strategy leverages the learned attention patterns to prioritize high-utility documents, eliminates the need for a fixed top-$K$ selection, and effectively filters out redundant or misleading content, thereby reducing unnecessary token consumption.

## 4 EXPERIMENT

### 4.1 EXPERIMENT SETUPS

**Dataset & metrics.** We evaluate our method on four widely used multi-hop QA datasets, including HotpotQA (Yang et al., 2018), 2WikiMultiHopQA (Ho et al., 2020) and MusiQue (Trivedi et al., 2022) and StrategyQA (Geva et al., 2021). Following standard RAG evaluation protocols (Yu et al., 2024), we use Exact Match (EM), F1 score (F1), and Accuracy (Acc) as our evaluation metrics.

**Baselines.** We compare PureCover with a broad range of baselines, including **dense retriever** e5-base-v2 (Wang et al., 2024), **relevance-oriented re-rankers**: bge-reranker-large (Xiao et al., 2024), Qwen3-Reranker-8B (Zhang et al., 2025), RankLlama (Ma et al., 2024), RankVicuna (Pradeep et al., 2023a), RankZephyr (Pradeep et al., 2023b), FIRST (Reddy et al., 2024), RankGPT (Sun et al., 2023), **coverage-oriented re-rankers**: IA-Select (Agrawal et al., 2009), MMR (Carbonell & Goldstein, 1998), xQuAD (Santos et al., 2010), and **re-rankers for RAG**: RADIO (Jia et al., 2024), SETR (Lee et al., 2025). More baseline details are provided in Appendix B.

**Implementation details.** We build PureCover on Qwen2.5-32B (Team, 2024) as the teacher and Qwen2.5-7B as the student model. For evaluation, we fix the retriever (*e5-base-v2*(Wang et al., 2024)) and generator (*Qwen2.5-7B-Instruct*). For a fair comparison, retrieval-only baselines retrieve the top-5 documents, while re-ranking methods operate under a re-ranking budget of 5, selecting

| Retrieval Method | Model | HotpotQA | | 2Wiki | | MusiQue | | StrategyQA | |
|---|---|---|---|---|---|---|---|---|---|
| | | EM | F1 | EM | F1 | EM | F1 | EM | Acc |
| | RETRIEVAL ONLY | | | | | | | | |
| | - | 31.60 | 41.68 | 28.20 | 35.14 | 10.20 | 18.22 | 66.40 | 67.60 |
| | CROSS-ENCODER RE-RANKER | | | | | | | | |
| | bge-reranker-large | 36.60 | 48.02 | 30.60 | 37.97 | 12.80 | 21.88 | 69.80 | 71.60 |
| | RADIO | 34.60 | 46.01 | 28.80 | 36.96 | 10.20 | 20.36 | 67.00 | 69.60 |
| | LLM-BASED RE-RANKER | | | | | | | | |
| | Qwen3-Reranker-8B | 36.60 | 47.45 | 29.00 | 36.23 | 13.20 | 21.91 | 69.60 | 71.40 |
| | RankVicuna | 36.60 | 46.88 | 28.60 | 35.86 | 13.00 | 21.95 | 70.00 | 71.60 |
| e5-base-v2 | RankLlama | 35.20 | 45.57 | 29.80 | 38.87 | 12.40 | 21.14 | 69.20 | 70.20 |
| | RankZephyr | 32.80 | 43.18 | 29.20 | 36.84 | 14.20 | _22.99_ | 68.80 | 71.00 |
| | FIRST | 35.80 | 47.40 | 29.00 | 36.54 | 12.80 | 21.86 | 68.60 | 70.40 |
| | ICR | 35.40 | 46.59 | 28.60 | 35.59 | 13.00 | 20.81 | 67.80 | 69.80 |
| | RankGPT(gpt-4o) | 36.80 | _48.05_ | _32.80_ | **41.03** | 13.80 | 22.86 | 69.40 | 71.20 |
| | COVERAGE-AWARE RE-RANKER | | | | | | | | |
| | IA-Select | 34.00 | 43.66 | 28.20 | 35.60 | 8.80 | 18.13 | 69.80 | 70.80 |
| | MMR | 35.60 | 47.34 | 30.60 | 37.99 | 13.00 | 22.00 | 68.20 | 70.80 |
| | xQuAD | _37.00_ | 48.01 | 31.80 | 39.55 | 12.60 | 20.52 | 68.20 | 70.20 |
| | SETR | 34.60 | 45.62 | 31.00 | 38.22 | _14.60_ | 22.61 | _70.20_ | _72.00_ |
| | OURS | | | | | | | | |
| | PureCover | **38.00** | **48.64** | **33.40** | _40.80_ | **14.80** | **23.79** | **70.60** | **72.20** |

Table 1: End-to-end question answering results across various re-ranking models. Each model applies re-ranking or selection over the top-20 documents retrieved using *e5-base-v2*. The **bold** and underlined indicate the best and second-best performances respectively.

| | HotpotQA | | 2WikiMultiHopQA | | MusiQue | | StrategyQA | |
|---|---|---|---|---|---|---|---|---|
| | EM | F1 | EM | F1 | EM | F1 | EM | Acc |
| PureCover (ours) | **38.00** | **48.64** | **33.40** | **40.80** | **14.80** | **23.79** | **70.60** | **72.20** |
| ABALTION MODELS | | | | | | | | |
| *w/o Attention* | 35.40 | 46.60 | 25.40 | 33.62 | 10.40 | 18.90 | _70.00_ | 71.00 |
| *w/o CoT weighting* | 35.80 | 47.03 | 31.00 | 38.23 | 13.40 | 22.07 | 69.60 | 70.40 |
| *w/o Position Bias Calibration* | 33.80 | 44.51 | 31.40 | 38.93 | 11.60 | 22.46 | 69.80 | _71.20_ |
| *w/o Optimization (Avg)* | _36.00_ | _46.87_ | 31.60 | 39.80 | _14.60_ | _23.66_ | 68.80 | 69.80 |
| *w/o Optimization (Max)* | 34.60 | 45.80 | _32.80_ | _40.78_ | 14.20 | 23.03 | 69.60 | _71.20_ |

Table 2: Ablation study of the proposed method, PureCover. The **bold** and underlined indicate the best and second-best performances, respectively.

these from the top-20 retrieved results. Additional details are in Appendix B. Our code is available at https://anonymous.4open.science/r/PureCover-2723.

## 4.2 EXPERIMENTAL RESULTS

We present our experimental results in Table 1. For coverage-oriented methods, we tune the trade-off coefficient $\lambda$ and report the best results. Based on these results, we have the following observations: our method consistently outperforms both relevance-oriented and coverage-aware baselines. Relevance-oriented re-rankers (e.g., LLM-based re-rankers) emphasize document-level relevance but overlook coverage, often failing to provide sufficient evidence for complex QA tasks. Although traditional coverage-aware methods improve list-level coverage, their diversity-oriented coverage often introduces unhelpful noise and leads to suboptimal performance. While coverage-oriented LLM re-rankers rely on the model's ability to select documents, they exhibit unstable performance on HotpotQA and 2Wiki. In contrast, our method explicitly balances coverage and noise via optimization, delivering substantial RAG improvements on complex queries.

## 4.3 FURTHER ANALYSIS

**Ablation study.** To assess the effectiveness of each component in PureCover, we conduct ablation studies on four multi-hop datasets (Table 2). We test the following variants: (1) *w/o Attention*: instead of using reasoning-attention signals to estimate $\mathbb{P}(d|e)$ (see Equation 7), we compute it based on the similarity between each CoT step and the document using BGE (Xiao et al., 2024); (2) *w/o CoT Weighting*: treat all information requirements as equally important, i.e., $\mathbb{P}(e|q) = 1$; (3) *w/o Position Bias Calibration*: use raw document–CoT attention weights without calibration; (4) *w/o Optimization*: replace the proposed greedy solver with simple heuristics—Top-$K$ by average atten-

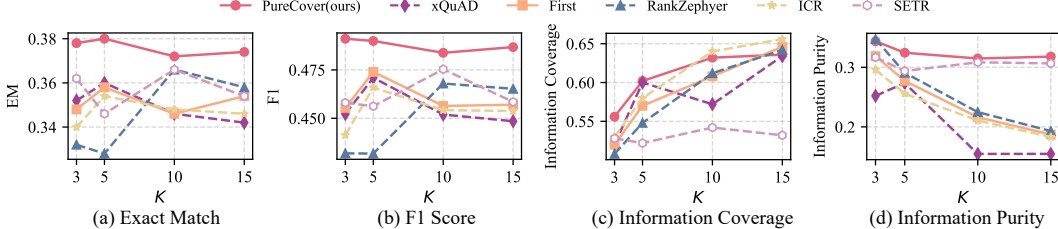

Figure 3: Performance of methods under different re-ranking budgets $K$ on HotpotQA dataset. The Information Coverage and Information Purity metrics measure the extent of coverage and noiselessness in the re-ranked set, with detailed formulas provided in Appendix B.

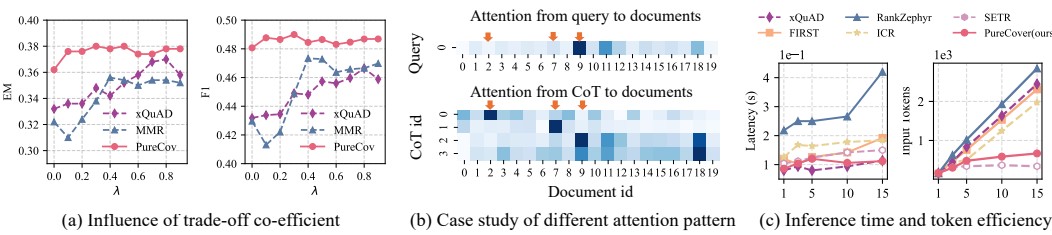

Figure 4: (a) Performance changes with different trade-off coefficients ($\lambda$); (b) Case study of attention patterns in identifying key evidence documents; (c) Inference and token efficiency.

tion (**Avg**) or by maximum single-step weight (**Max**). Results show that removing any component consistently harms performance: ignoring position bias calibration misestimates document importance, while dropping the solver yields incomplete evidence coverage and degrades RAG accuracy.

**Can PureCover balance coverage and noise under varying re-ranking budget $K$?** Beyond EM and F1 scores, we introduce *Information Coverage* and *Information Purity* metrics to measure the presence of key evidence and the noise in the document set, respectively. As shown in Figure 3, our method PureCover achieves high coverage of key information even with small $K$ (e.g., $K = 3$), while effectively suppressing unhelpful noise even with large $K$ (e.g., $K = 15$). Traditional top-$K$ re-rankers show a significant increase in noise (decrease in information purity) as the budget increases, which compromises RAG performance (EM) despite high coverage. In contrast, while SETR can select a relatively clean (unnoisy) set under large budgets, its low information coverage limits the generator's ability to answer complex questions, resulting in suboptimal RAG performance. Our method effectively balances these two critical factors.

**Influence of trade-off coefficient $\lambda$.** A sensitivity analysis of the noise–coverage coefficient $\lambda$ compares our method with traditional coverage-aware re-ranker (i.e., MMR and xQuAD) on HotpotQA. A smaller $\lambda$ prioritizes coverage, while a larger $\lambda$ emphasizes relevance (noiselessness). Figure 4 (a) illustrates that traditional baselines perform poorly at low $\lambda$ (high coverage), as their diversity-based coverage introduces excessive noise that hurts accuracy. In contrast, our method is more robust and stable, effectively balancing coverage and noise and maintaining strong performance across various $\lambda$ settings. We can also observe the impact of noise on our method: prioritizing either full coverage ($\lambda$=0, EM=0.362) or full noiselessness ($\lambda$=1, EM=0.372) does not yield optimal results. Instead, balancing these two objectives gives our method its best performance ($\lambda$=0.3, EM=0.38, F1=0.490), demonstrating its ability to balance noise and coverage.

**Effectiveness of attention pattern during reasoning.** To validate the effectiveness of our method, which uses attention from goal-oriented reasoning steps, we visualize its attention heatmap and compare it to a query-based attention baseline (i.e, ICR). As shown in Figure 4(b), the baseline method shows poor coverage of key documents, often getting distracted by noise (e.g., Documents 11 and 18). This limited focus hinders its ability to adequately cover all relevant information for complex queries. In contrast, our method aggregates attention across different CoT reasoning steps, successfully focusing on key documents. This significantly improves the coverage of the re-ranked list, ensuring all necessary information is considered to answer the query effectively.

**Inference time and token efficiency.** We conducted experiments to evaluate the superiority of our method's inference efficiency and LLM generator token costs. As shown in Figure 4 (c), point-wise methods like RankZephyr exhibit a significant increase in inference time and token costs as the re-ranking length budget $K$ grows from 1 to 15. In contrast, our method not only maintains comparable inference efficiency to traditional IR methods (i.e., xQuAD) but also achieves substantial savings in input token costs for the RAG generator component.

**Other experimental analysis.** Due to space constraints, more experiments of alternative retrieval methods, teacher models, re-ranking budget $K$, and case studies are deferred to Appendix C.

## 5 RELATED WORK

**RAG.** RAG has become a crucial technique for mitigating hallucinations and improving factual accuracy in LLMs. Most studies focused on optimizing the integration of LLMs with retrieval modules (Gao et al., 2023; Zhao et al., 2023). For example, some work has explored dynamic retrieval to determine if and when retrieval is needed (Jeong et al., 2024). Others have tackled complex questions by iteratively decomposing them (Kim et al., 2023; Sarthi et al., 2024). Another line of research focused on developing IR models better suited for RAG, such as using PPO (Schulman et al., 2017) to train a document selector (Ke et al., 2024) or building a cross-encoder re-ranker to find documents aligned with the query's rationale (Jia et al., 2024). Despite these advances, most of these methods prioritize relevance over a crucial list-level coverage (Xie et al., 2024; Es et al., 2024), which significantly limits their effectiveness in answering complex, multi-hop queries.

**Coverage-aware Re-ranking.** Coverage-aware re-ranking, a well-established area in traditional IR, aims to ensure ranked results span diverse subtopics or intents (Wu et al., 2024; Clarke et al., 2008). Heuristic methods like MMR (Carbonell & Goldstein, 1998) and xQuAD (Santos et al., 2010) greedily select documents that are complementary to those already chosen. While non-heuristic learning-to-rank approaches can directly optimize coverage metrics (Yan et al., 2021), they are often limited by the high cost of human annotations. A few recent works highlight the importance of coverage in RAG (Es et al., 2024; Xie et al., 2024; Lee et al., 2025). For example, SETR (Lee et al., 2025) directly distilled the coverage ability from teacher models. However, previous studies have emphasized that increasing coverage inevitably introduces noise (Chen et al., 2023). Existing methods overlook this factor, preventing them from maintaining strong RAG performance under large re-ranking budgets.

**LLM as Re-ranker.** LLMs exhibit strong zero-shot ranking abilities due to their vast world knowledge, leading to the development of various ranking methods: point-wise (Zhang et al., 2025; Ma et al., 2024; Sun et al., 2023; Pradeep et al., 2023a), pair-wise (Qin et al., 2023), and list-wise (Reddy et al., 2024; Chen et al., 2024). Recent works have introduced faster decoding strategies (Reddy et al., 2024) or used calibrated attention for ranking (Chen et al., 2024). However, all of these LLM-based re-rankers follow the traditional IR Top-$K$ re-ranking paradigm, aiming to rank the most query-relevant documents at the top. This approach overlooks crucial factors like coverage and noise, makes them difficult to apply effectively in RAG scenarios.

## 6 CONCLUSION

This work identifies and addresses a key limitation of existing RAG systems: the direct application of re-ranking methods designed for human users to LLMs. We highlight that this direct adoption overlooks a crucial gap: unlike humans, LLMs lack robust selective attention to filter noise and focus on key evidence. To bridge this gap, we introduce PureCover, a novel document selection framework tailored for RAG. Moving beyond the limitations of traditional Top-K re-ranking, PureCover formulates document selection as a multi-objective optimization problem, dynamically balancing information coverage and noise minimization. Our approach uniquely leverages the internal attention patterns of LLMs during goal-oriented reasoning to precisely identify and select key evidence. For practical deployment, we developed an efficient set-wise distillation strategy to transfer this sophisticated selection capability to compact LLM selectors. Extensive experiments on four multi-hop QA benchmarks reveals that PureCover consistently outperforms state-of-the-art baselines, demonstrating its ability to achieve a superior balance of coverage and noise, thereby significantly enhancing RAG performance on complex tasks.

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

## APPENDIX

## A GREEDY ALGORITHM

---

**Algorithm 1:** Greedy Algorithm

**Input:** Document-CoT overage matrix $\mathbf{W} \in [0,1]^{N \times K_1}$, CoT importance vector $\mathbf{e} \in [0,1]^N$, re-ranking list length budget $K_2$, coverage-relevance trade-off weight $\lambda$, truncation threshold $\tau$.

**Output:** Selected document set $S$.

1: Initialize selected document set $S = \varnothing$, remaining document set $\mathcal{R} = \mathcal{D}_{\text{retrieve}}$, evidence coverage vector $\mathbf{p} = 0^{K_1}$.

2: **while** $|S| < K_2$ **and** $g^* > \tau$ **do**

3:   $g^* = -\infty, d^* = \varnothing$

4:   **for** $d \in \mathcal{R}$ **do**

5:     $\mathbf{p}' = \mathbf{e} \odot [1 - (1 - \mathbf{p}) \odot (1 - \mathbf{W}_d)]$ //Compute Coverage Vector

6:     $\Delta_g = \sum(\mathbf{p}' - \mathbf{p})$

7:     $g = \Delta_g - \lambda \cdot [1 - \max(\mathbf{W}_d \odot \mathbf{e})]$ //Compute Noise Vector

8:     **if** $g > g^*$ **then**

9:       $d^* \leftarrow d, g^* \leftarrow g, \mathbf{p}^\star \leftarrow \mathbf{p}'$

10:      **end if**

11:   **end for**

12:   $S \leftarrow S \cup \{d^*\}, \mathcal{R} \leftarrow \mathcal{R} \setminus \{d^*\}$ //Update Selection Set

13:   $\mathbf{p} \leftarrow \mathbf{p}^\star$

14: **end while**

15: **return** $S$

---

# B    ADDITIONAL EXPERIMENTAL DETAILS

## B.1    IMPLEMTATION DETAILS.

We implemented all baseline methods using the FlashRAG (Jin et al., 2024) and Rankify (Abdallah et al., 2025) library. Following previous works (Sun et al., 2025), we randomly select 500 samples from the test sets of each dataset as our final test set for all baselines and our method. We set the training epochs to 10 with a learning rate of 1e-5. For the temperature coefficients $\tau_1$ and $\tau_2$, we set then both to be 4e-5. All experiments are conducted on four NVIDIA RTX A6000 48G GPUs.

## B.2    METRICS

In addition to evaluating the performance of our document selector based on the answer quality of the downstream LLM generator using metrics like EM and F1, we further define new metrics based on recall and precision to assess how well the selector balances coverage and noise under different re-ranking list budget $K$. Before presenting the formal definitions, we first explain how we handle the re-ranking list length budget $K$. After the document selector (re-ranker) outputs the selected documents $\mathcal{D}_{\text{rerank}}$, the RAG system typically constrains the set to size K to comply with the input token limitations of the LLM generator (Lee et al., 2025). Specifically, $\mathcal{D}_{\text{rerank}}^{(K)}$ refers to the subset of documents selected under budget K (i.e., the top-K documents from $\mathcal{D}_{\text{rerank}}$), which can be formally defined as:

$$\mathcal{D}_{\text{rerank}}^{(K)} = \begin{cases} \mathcal{D}_{\text{rerank}}, & \text{if } |\mathcal{D}_{\text{rerank}}| \leq K, \\ \text{Top-}K(\mathcal{D}_{\text{rerank}}), & \text{if } |\mathcal{D}_{\text{rerank}}| > K, \end{cases} \tag{9}$$

where Top-$K(\mathcal{D}_{\text{rerank}})$ denotes the top $K$ documents in $\mathcal{D}_{\text{rerank}}$ ranked by their re-rannking scores.

**Information Coverage** measures how well the selected document set under a budget $K$ covers the necessary ground-truth evidence. Formally, it is defined as the fraction of gold documents (i.e., documents that contain the golden answer) that appear in the top-$K$ re-ranked list.

$$\text{Information Coverage@}K = \frac{|\mathcal{D}_{\text{rerank}}^{(K)} \cap \mathcal{D}_{\text{gold}}|}{|\mathcal{D}_{\text{gold}}|} \tag{10}$$

where $\mathcal{D}_{\text{gold}}$ is the set of all documents that contain the golden answer. This metric effectively measures whether the selected documents cover all the information required to answer the question. A higher value indicates that the selected documents achieve greater coverage.

**Information Purity** evaluates the proportion of useful documents within the selected set under a budget $K$. Specifically, it is defined as the fraction of gold documents among the top-$K$ re-ranked documents.

$$\text{Information Purity@}K = \frac{|\mathcal{D}_{\text{rerank}}^{(K)} \cap \mathcal{D}_{\text{gold}}|}{|\mathcal{D}_{\text{rerank}}^{(K)}|} \tag{11}$$

where $\mathcal{D}_{\text{gold}}$ is the set of all documents that contain the golden answer. This metric effectively measures the amount of irrelevant noise present in the selected documents. A higher value indicates that the selected documents contain less noise.

Since many RAG datasets do not provide annotations for gold documents (Jin et al., 2024; Yang et al., 2018; Ho et al., 2020), we follow prior work and use $\tilde{\mathcal{D}}_{\text{gold}}$ to approximate $\mathcal{D}_{\text{gold}}$. Specifically, $\tilde{\mathcal{D}}_{\text{gold}} = \{d_i \in \mathcal{D} \mid d_i \text{ contains } a_{\text{gold}}\}$, i.e., the set of documents that include the gold answer. Previous studies (Jin et al., 2024) have shown that this approach provides a reasonable estimate of a re-ranker's performance.

## B.3    DATASETS

The experiments were conducted on the following four benchmark datasets:

• **HotpotQA** (Yang et al., 2018) is a large-scale multi-hop QA dataset containing 113k question–answer pairs derived from Wikipedia. Each question requires reasoning across multiple documents, with sentence-level supporting facts annotated. The dataset features diverse query types,

including comparison questions, and emphasizes compositional reasoning and explainability, making it a widely used benchmark for multi-hop retrieval and reasoning systems.

- **2WikiMultiHopQA** (Ho et al., 2020) is a multi-hop QA dataset that integrates Wikipedia text with Wikidata triples to assess step-by-step reasoning. Each question is paired with an explicit reasoning path connecting entities across documents. The dataset evaluates compositional reasoning and requires models to leverage both unstructured text and structured knowledge for accurate answers.

- **MusiQue** (Trivedi et al., 2022) is a multi-hop QA benchmark that prevents reasoning shortcuts by requiring genuine multi-step reasoning. It contains about 25k questions constructed from linked single-hop queries, each involving 2–4 reasoning steps. The dataset emphasizes strong logical dependencies between steps and includes unanswerable variants to test robustness, making it more challenging than earlier benchmarks and highlighting a larger performance gap between humans and models.

- **StrategyQA** (Geva et al., 2021) is a multi-hop QA benchmark designed to evaluate implicit reasoning. Each question is a yes/no query requiring models to infer a series of hidden reasoning steps, with supporting evidence drawn from relevant Wikipedia paragraphs. The dataset contains 2,780 examples covering a wide range of topics and reasoning strategies, emphasizing compositional reasoning and strategy inference. Compared to humans, current models still show a substantial performance gap, highlighting the challenge of implicit multi-step reasoning.

## B.4 BASELINES

We consider the following re-ranking models as baselines. They include relevance-oriented cross-encoder and LLM-based re-rankers, traditional coverage-oriented models, and a LLM-based coverage-aware re-ranker.

- **bge-reranker-large** (Xiao et al., 2024) is a lightweight cross-encoder from BAAI that scores query–passage pairs using full cross-attention, providing more accurate relevance judgments than embedding-based models. Fine-tuned on large-scale data, it is widely used to rescore top-$k$ retrieval results. As a strong open-source baseline, it represents state-of-the-art conventional re-ranking focused on individual document relevance.

- **Qwen3-Reranker** (Zhang et al., 2025) is part of the Qwen3 Embedding series, built on the Qwen3 foundation models. The re-ranker is point-wise: it scores each document independently using the yes/no logits given a query, rather than comparing pairs or lists. It is supervised fine-tuned (SFT) on high-quality labeled data to optimize relevance ranking performance. The Qwen3-Reranker models come in different sizes (0.6B, 4B, 8B parameters) to trade off efficiency and accuracy, and have shown strong results across multilingual and retrieval benchmarks. In this paper, we use Qwen3-Reranker-8B as the baseline for comparison [1].

- **RankLlama** (Ma et al., 2024) is a pointwise re-ranker built on the LLaMA-2-7B model. Given a query and a candidate passage, it produces a relevance score to reorder retrieved documents. It achieves strong performance in both in-domain and zero-shot settings, serving as a competitive open-source baseline for document re-ranking.

- **RankVicuna**(Pradeep et al., 2023a) is a 7B open-source listwise re-ranker built on the Vicuna-7B model. It takes a query and a list of passages as input and outputs a ranked list of passage indices. Trained with GPT-generated supervision, it achieves performance comparable to GPT-3.5 on benchmarks such as TREC DL, offering a transparent alternative to proprietary re-rankers.

- **RankZephyr** (Pradeep et al., 2023b) is a zero-shot list-wise re-ranker built on the Zephyr-7B model. Fine-tuned with GPT-4-generated rankings, it produces ordered lists of passage indices given a query and candidate passages. It achieves performance comparable to GPT-4, surpassing it on some benchmarks. Its open-source and reproducible design makes it a robust baseline for evaluating listwise re-ranking methods.

- **FIRST**(Reddy et al., 2024) is a zero-shot listwise re-ranker that frames ranking as a single-token decoding task, enabling fast and efficient passage selection. Despite its simplicity, it achieves com-

---

[1] https://huggingface.co/Qwen/Qwen3-Reranker-8B

petitive performance and serves as a strong open-source baseline for evaluating the raw ranking ability of instruction-tuned LLMs. In this paper, we use the Zephyr-7B–based open-source version[2].

• **ICR** (Chen et al., 2024) is a zero-shot re-ranker that leverages the attention patterns of large language models to estimate document relevance without generation. Given a query and candidate passages, it extracts attention-based scores to rank documents efficiently. ICR achieves competitive performance with minimal computation, providing a fast, open-source alternative for zero-shot document re-ranking.

• **RankGPT** (Sun et al., 2023) is a GPT-4–based re-ranker accessed via OpenAI's API, operating in a zero-shot setting to rank passages given a query. It achieves state-of-the-art performance but is closed-source, non-reproducible, and costly. RankGPT4 serves as an upper-bound baseline for evaluating the effectiveness of our approach against the strongest proprietary re-ranker. We use `gpt-4o`[3] from OpenAI to run RankGPT.

The following are the coverage-aware baselines. Since labeled or pre-collected query subtopics are often unavailable in RAG datasets, we use GPT-4o to identify the information requirements within each query and treat them as subtopics.

• **IA-Select** (Agrawal et al., 2009) is one of the most widely used coverage-aware algorithms that attempts to maximize the probability that a user finds at least one useful result within $k$ results. Given a set of candidate documents and a set of subtopics or intents, IA-Select greedily selects documents that provide the highest expected coverage over these subtopics.

• **xQuAD** (Santos et al., 2010) is a widely used coverage-aware re-ranking algorithm that extends IA-Select by explicitly balancing relevance and coverage. It iteratively selects documents to maximize coverage over multiple query aspects or subtopics while also considering individual document relevance.

• **Maximize Marginal Relevance (MMR)** (Carbonell & Goldstein, 1998) is a widely used coverage-aware re-ranking algorithm that balances relevance and novelty in document retrieval. It iteratively selects documents to maximize relevance to the query while minimizing redundancy with already selected documents. This approach ensures that the final set of documents provides comprehensive coverage of the query's information needs without unnecessary repetition.

• **SETR** (Lee et al., 2025) is a set-wise re-ranker for RAG that models coverage by decomposing a query into information requirements using Chain-of-Thought reasoning and selecting a set of documents that collectively satisfy them. Different from the traditional Top-$K$ re-ranking strategy, SETR distills the ability and knowledge from a powerful GPT-4o teacher into a smaller student model. Following the authors' implementation, we utilize GPT-4o as the teacher model and Qwen2.5-7B as the student model, which is the same as our method, PureCover.

• **RADIO** (Jia et al., 2024) is a cross-encoder-based re-ranker for bridging the gap between retrievers and generators in RAG systems. It addresses the mismatch where retrievers select documents that may not fully support the reasoning needs of LLMs. RADIO extracts reasoning rationales from LLM outputs and uses them to guide re-ranking and retriever fine-tuning, aligning retrieval with generation needs. We utilize bge-reranker-large (Xiao et al., 2024) as the re-ranking model for RADIO.

## C  ADDITIONAL EXPERIMENT

### C.1  EXPERIMENT ON OTHER RETRIEVAL MODEL

We further evaluate our method with BM25 as the retrieval backbone to test its robustness under different retrieved document distributions. Table 3 reports the results. Across all multi-hop QA datasets, our approach consistently achieves the best performance compared to baselines, except on StrategyQA, where it performs comparably to SETR. Notably, traditional coverage-aware methods degrade significantly on more complex datasets such as MuSiQue and StrategyQA, indicating their limited ability to handle challenging reasoning scenarios. While the LLM-based selector SETR

---

[2]https://huggingface.co/rryisthebest/First_Model

[3]`gpt-4o` in this paper refers to `gpt-4o-2024-11-20` from OpenAI

| Retrieval Method | Model | HotpotQA | | 2Wiki | | MusiQue | | StrategyQA | |
|---|---|---|---|---|---|---|---|---|---|
| | | EM | F1 | EM | F1 | EM | F1 | EM | Acc |
| | RETRIEVAL ONLY | | | | | | | | |
| | - | 30.80 | 40.63 | 28.40 | 34.08 | 7.40 | 16.12 | 62.20 | 65.40 |
| | CROSS-ENCODER RE-RANKER | | | | | | | | |
| | Bge-Reranker-Large | 36.60 | 47.22 | 32.60 | 39.52 | 10.20 | 18.93 | 65.00 | 66.00 |
| | RADIO | 34.60 | 46.01 | 28.80 | 36.96 | 10.20 | 20.39 | 67.00 | 69.60 |
| | LLM-BASED RE-RANKER | | | | | | | | |
| | Qwen3-Reranker-8B | 34.40 | 46.19 | 29.60 | 36.18 | 9.80 | 18.75 | 65.40 | 67.20 |
| | RankVicuna | 34.00 | 44.69 | 27.40 | 33.82 | 9.40 | 18.27 | 64.60 | 66.60 |
| BM25 | RankLlama | 34.40 | 46.06 | 28.40 | 34.97 | 9.80 | 18.74 | 64.60 | 66.80 |
| | RankZephyr | 33.20 | 44.93 | 30.80 | 37.28 | 11.20 | 19.77 | 64.00 | 65.80 |
| | FIRST | 33.40 | 45.08 | 30.40 | 37.26 | 11.00 | 19.34 | 64.20 | 67.40 |
| | ICR | 35.20 | 46.22 | 31.00 | 37.45 | 8.80 | 17.15 | 62.40 | 65.00 |
| | RankGPT (gpt-4o) | 35.00 | 46.49 | 33.20 | 39.89 | 10.80 | 19.04 | 64.60 | 66.80 |
| | COVERAGE-AWARE RE-RANKER | | | | | | | | |
| | IA-Select | 34.00 | 43.66 | 28.20 | 35.62 | 8.60 | 17.92 | 60.40 | 63.80 |
| | MMR | 34.60 | 45.75 | 31.00 | 37.52 | 11.00 | 19.69 | 62.20 | 64.20 |
| | xQuAD | 34.20 | 45.07 | 28.60 | 34.96 | 8.40 | 17.45 | 62.20 | 64.40 |
| | SETR | 36.80 | 47.36 | 31.20 | 38.36 | 13.80 | 22.65 | **70.00** | **71.80** |
| | OURS | | | | | | | | |
| | PureCover | **37.80** | **47.74** | **34.00** | **40.30** | **15.20** | **23.39** | 69.60 | 68.00 |

Table 3: End-to-end question answering results across various ranking models. Each model applies re-ranking or selection over the top-20 documents retrieved using *BM25*. The **bold** and underlined indicate the best and second-best performances respectively.

surpasses relevance-oriented baselines on several datasets, its gains are unstable and it performs poorly on 2WikiMultiHopQA. In contrast, our method demonstrates stable and superior improvements across all datasets.

## C.2 EXPERIMENT ON DIFFERENT RE-RANKING BUDGET $K$

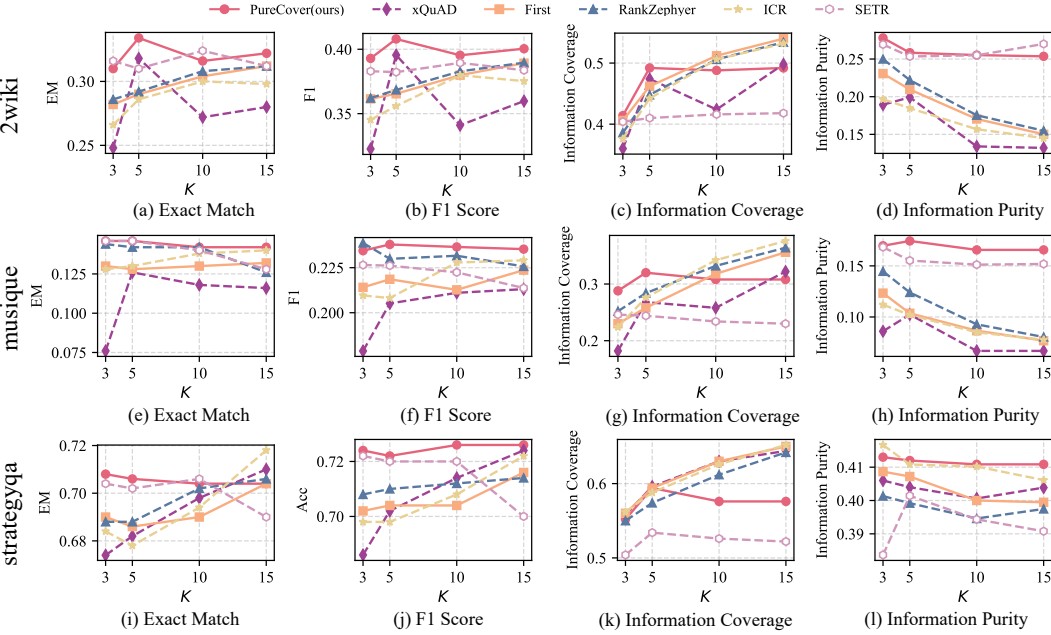

Figure 5: Performance of PureCover under varying re-ranking budgets $K$. Information Coverage and Information Purity are evaluated with respect to documents containing the gold answer. Subfigures (a–d), (e–h), and (i–l) correspond to experiments on 2WikiMultihopQA, Musique, and StrategyQA, respectively.

As shown in Figure 5, we evaluate the performance of different methods on 2WikiMultihopQA, Musique, and StrategyQA under varying re-ranking list budgets $K$.

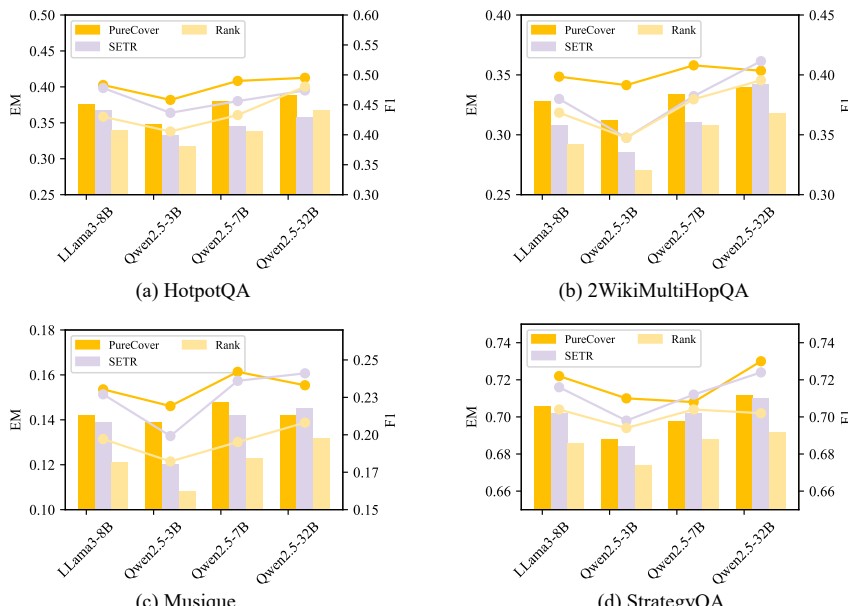

Figure 6: Ablation study on different teacher models to isolate method-level effects. The student model is applied using Qwen2.5-7B.

Overall, our method consistently achieves strong and stable results across all three datasets, maintaining high end-to-end RAG performance regardless of the budget $K$. In contrast, traditional Top-$K$ re-ranking strategies (e.g., xQuAD, First, RankZephyr) tend to perform poorly when $K$ is small (e.g., $K = 3$). This is because they struggle to ensure coverage of the critical evidence (see Figures (c), (g), and (k)). Whether based on LLMs or cross-encoders, conventional IR re-rankers suffer from insufficient coverage under low $K$. As $K$ increases, these methods gradually improve coverage by leveraging relevance, but this improvement comes at the cost of introducing significant noise, which ultimately limits RAG performance. By contrast, our method is able to guarantee sufficient coverage at small $K$ and suppress noise growth at large $K$, thereby sustaining relatively high performance across different budgets. It is also worth noting that SETR does not rely on a fixed Top-$K$ list, which allows it to achieve strong noise control (e.g., in Figures (d) and (h), its information purity reaches 0.270 and 0.152 at $K = 15$). However, due to its relatively low coverage, SETR falls short of achieving optimal RAG performance.

## C.3 FAIR COMPARISON ON SAME STUDENT MODEL

To ensure a fair comparison and verify that our multi-objective training approach offers methodological advantages over existing re-ranking strategies, we conduct experiments on the same student model (Qwen2.5-7B) while varying the teacher models. Specifically, we use models of different sizes and families, including LLaMA3-8B, Qwen2.5-3B, Qwen2.5-7B, and Qwen2.5-32B, as teachers. We compare our method PureCover with two strong alternatives: (1) SETR (Lee et al., 2025), which identifies query information requirements and performs document selection with the teacher model, and (2) list-wise ranking with LLMs (Rank) (Pradeep et al., 2023b). In this setting, the re-ranking budget $K$ is fixed at 5.

Experimental results in Figure 6 show that the chosen baseline methods relying on teacher model inference for document ranking are highly dependent on model capacity and size. For example, both list-wise Rank and SETR perform poorly with smaller teachers (e.g., 3B) but significantly better with larger ones (e.g., 30B). In contrast, our approach, by combining attention with optimization rather than relying only on raw model capacity, compensates for the weaker reasoning abilities of smaller models and thus achieves competitive performance even with lightweight teachers.

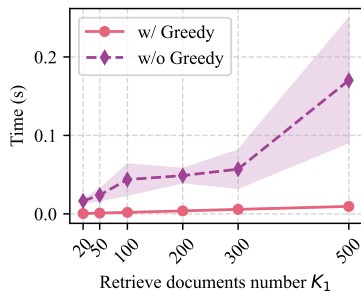
(a) Training time w.r.t retrieve document number

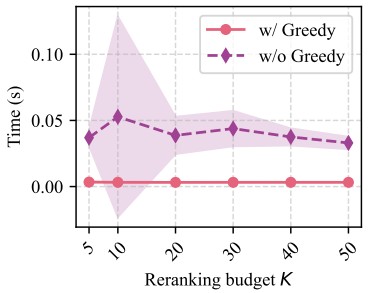
(a) Training time w.r.t reranking budget

Figure 7: Training efficiency analysis under different settings: (a) number of retrieved documents $K_1$, and (b) re-ranking budget $K$. When varying $K_1$, we fix the re-ranking budget to $K = 10$; when varying $K$, we fix the number of retrieved documents to $K_1 = 100$. Each experiment is repeated 20 times, and we report the mean and variance.

## C.4 TRAINING EFFICIENCY ANALYSIS

We propose a greedy algorithm to efficiently solve the multi-objective document selection problem, which greatly improves the training efficiency of our framework. To evaluate its effectiveness, we conduct experiments under varying numbers of retrieved documents $K_1$ and different re-ranking budgets $K$. Specifically, when analyzing the impact of $K_1$, we fix the re-ranking budget to $K = 10$; when analyzing the impact of $K$, we set the number of retrieved documents to $K_1 = 100$. We use the **ECOS-BB** (Domahidi et al., 2013) solver the solve the original binary optimization problem in Equation 8.

Figure 7 reports the time cost of our method with and without the greedy algorithm under these two settings. For a fair comparison, we exclude the time for LLM reasoning and attention extraction, and only measure the efficiency of the optimization module. The results show that the greedy algorithm significantly accelerates training, especially when the number of retrieved documents $K_1$ is large. Moreover, regardless of the re-ranking budget $K$, our greedy-based optimizer requires nearly zero training time, while the version without greedy optimization takes 1–2 seconds per iteration. These results demonstrate that the proposed greedy algorithm not only improves scalability to large retrieval sizes but also ensures consistently efficient training across different re-ranking budgets.

## C.5 CASE STUDY

To further illustrate the benefits of our proposed method, we present example cases from the HotpotQA dataset. Figure 8 shows the Top-3 retrieved documents ranked by our method, the best-performing coverage-aware baseline (xQuAD), and the LLM-based re-ranker (RankZephyr). The example query is a multi-hop question that requires multiple evidence documents as input to the generator in order to be answered correctly. For clarity, we highlight different aspects of the query and documents in different colors. And the noisy information is highlighted using the red color.

We observe that our method successfully ranks documents by considering both coverage and noise control. Specifically, the Top-3 results each capture a distinct aspect of the query, ensuring comprehensive evidence for reasoning while avoiding irrelevant content. In contrast, xQuAD achieves diversity by selecting documents about different subtopics, but these subtopics are weakly related to the actual CoT steps and often introduce noise that distracts the generator. RankZephyr, on the other hand, places the most relevant documents at the top, but due to the lack of coverage awareness, it produces a redundant list (e.g., two documents both describing NBA players), with some documents even being noisy (the document about T.R. Dunn), which ultimately fails to support the multi-hop reasoning process and distracting the LLM generator.

**Example Query1**: What distinction is held by the former NBA player who was a member of the Charlotte Hornets during their 1992-93 season and was head coach for the WNBA team Charlotte Sting?

**Document Reranked by PureCover:** 1. Muggsy Bogues Tyrone Curtis "Muggsy" Bogues (born January 9, 1965) is an American retired basketball player. The shortest player ever to play in the National Basketball Association. 2. During the season, the Sting traded veteran Dawn Staley to the Houston Comets and named Charlotte basketball icon Muggsy Bogues as their new head coach late in the season. 3. In the 1992 NBA draft, the Hornets selected center Alonzo Mourning out of Georgetown with the second overall pick. With the addition of Mourning, along with second-year star Larry Johnson and Muggsy Bogues

**Response Given by PureCover:** shortest player ever to play in the National Basketball Association ✅

**Document Reranked by xQuAD:** 1. Anne Donovan. She then led the Charlotte Sting to the WNBA Finals in 2001, losing to the Los Angeles Sparks. 2. Charlotte Sting. Michael Jordan renamed the Bobcats. Uniforms: The Charlotte Sting was one of the eight original WNBA franchises that began play in 1997. 3. "1992-93 Charlotte Hornets season" season, Kendall Gill was traded to the Seattle SuperSonics. Signed LaMark Baker as a free agent. Signed Lorenzo Williams as a free agent.

**Response Given by xQuAD:** Anne Donovan ❌

**Document Reranked by RankZephyr:** 1. Muggsy Bogues. Muggsy Bogues Tyrone Curtis "Muggsy" Bogues (born January 9, 1965) is an American retired basketball player. The shortest player ever to play in the National Basketball Association. 2. "T. R. Dunn" his career, and he was widely regarded as one of the best rebounding guards of the 1980s. 3. "Charlotte Sting" to build for the future -trading with the Sacramento Monarchs for Tangela Smith and a second-round draft pick in the 2006 draft in a deal that saw Nicole Powell traded to Sacramento.

**Response Given by RankZephyr:** Muggsy Bogues ❌

Figure 8: Case study on HotpotQA. The query contains multiple information needs, highlighted with different background colors. The red color denotes the noise information. We compare our method against the best-performed LLM-based and coverage-aware baselines (xQuAD, RankZephyr).

## D  COMPUTATION OF TOKEN-LEVEL ATTENTION SCORE

In the Transformer architecture (Vaswani et al., 2017), the attention mechanism operates on three matrices: queries $\mathbf{Q} \in \mathbb{R}^{m \times d_k}$, keys, $\mathbf{K} \in \mathbb{R}^{n \times d_k}$, and values $\mathbf{V} \in \mathbb{R}^{n \times d_v}$. The attention function is defined as:

$$\text{Attention}(\mathbf{Q}, \mathbf{K}, \mathbf{V}) = \text{softmax}\left(\frac{\mathbf{Q}\mathbf{K}^\top}{\sqrt{d_k}}\right)\mathbf{V}. \tag{12}$$

Here, each element $(QK^\top)_{ij} = q_i \cdot k_j$ measures the similarity between the $i$-th query and the $j$-th key. Dividing by $\sqrt{d_k}$ controls the magnitude of the dot products, preventing overly large values that could destabilize training. The softmax function is applied row-wise, normalizing the scores into a probability distribution whose weights sum to one. These attention weights are then used to compute a weighted sum of the value vectors $V$, producing the output representation.

At the token level, the attention weight

$$\mathbf{A}_{ij} = \text{softmax}\left(\frac{q_i \cdot k_j}{\sqrt{d_k}}\right)$$

represents how much token $i$ attends to token $j$. These weights are computed dynamically for each input, enabling the model to capture semantic dependencies between tokens.

## E  PROOFS OF THEOREM 1 AND PROPOSITION 1

*Proof of Theorem 1.* According to previous studies of multi-objective optimization (Deb et al., 2016), the problem of balancing competing objectives can be expressed in the weighted sum form:

$$\max_{\mathcal{D}_{\text{rerank}} \subseteq \mathcal{D}_{\text{retrieve}}, |\mathcal{D}_{\text{rerank}}| \leq K} \text{Cover}(\mathcal{D}_{\text{rerank}}) - \lambda \cdot \text{Noise}(\mathcal{D}_{\text{rerank}}),$$

where $\lambda$ is a trade-off coefficient between the two objectives.

**Step 1: Coverage function.**  From Equation 4, the coverage of the selected set is:

$$\text{Cover}(\mathcal{D}_{\text{rerank}}) = \sum_{i=1}^{|\mathcal{E}|} \mathbf{e}_i \cdot \mathbb{P}(e_i \text{ covered by } \mathcal{D}_{\text{rerank}}).$$

For each information requirement $e_i$, the probability it is not covered by any selected document is:

$$\prod_{d \in \mathcal{D}_{\text{rerank}}} (1 - \mathbf{W}_{d,i}).$$

Thus, the probability it is covered is:

$$1 - \prod_{d \in \mathcal{D}_{\text{rerank}}} (1 - \mathbf{W}_{d,i}).$$

Introducing the binary selection variable $\mathbf{x}_d \in \{0, 1\}$, where $\mathbf{x}_d = 1$ if document $d$ is selected, the above product becomes:

$$\prod_{d=1}^{K_1} (1 - \mathbf{W}_{d,i}\mathbf{x}_d).$$

Therefore, the coverage term is:

$$\sum_{i=1}^{|\mathcal{E}|} \mathbf{e}_i \left[ 1 - \prod_{d=1}^{K_1} (1 - \mathbf{W}_{d,i}\mathbf{x}_d) \right].$$

**Step 2: Noise function.**  From Equation 5, noise is defined as:

$$\text{Noise}(\mathcal{D}_{\text{rerank}}) = \sum_{d \in \mathcal{D}_{\text{rerank}}} \text{DisUtil}(d|q).$$

Here, disutility for a document is defined as:

$$\text{DisUtil}(d|q) = 1 - \max_i \mathbf{W}_{d,i}\mathbf{e}_i.$$

Using $\mathbf{x}_d$ to indicate selection, the noise term becomes:

$$\sum_{j=1}^{K_1} (1 - \max(\mathbf{W}_j \odot \mathbf{e})) \, \mathbf{x}_j,$$

where $\odot$ denotes element-wise multiplication.

**Step 3: Binary optimization formulation.**  Combining the coverage and noise terms, and introducing the budget constraint $\sum_i \mathbf{x}_i \leq K$, we obtain:

$$\max_{\mathbf{x}} \sum_{i=1}^{|\mathcal{E}|} \mathbf{e}_i \left[ 1 - \prod_{d=1}^{K_1} (1 - \mathbf{W}_{d,i}\mathbf{x}_d) \right] - \lambda \sum_{j=1}^{K_1} (1 - \max(\mathbf{W}_j \odot \mathbf{e})) \, \mathbf{x}_j,$$

$$\text{s.t.} \ \sum_i \mathbf{x}_i \leq K, \quad \mathbf{x}_i \in \{0, 1\}, \ \forall i.$$

This exactly matches the statement in Equation 8, proving that the denoised coverage-aware document selection problem can indeed be formulated as a 0–1 integer (binary) multi-objective optimization problem. $\square$

*Proof of Proposition1.*  We prove the claim by showing each per-evidence term is monotone and submodular, and then use closure properties of submodular functions. Let's define the multi-objective objective function in Problem 8 as $g(S) = G(S) - \lambda \cdot R(S)$.

**(1) Per-evidence term.** Fix an evidence $e_i$. Define

$$h_i(S) \ = \ 1 - \prod_{d \in \mathcal{S}} (1 - \mathbf{W}_{d,i}), \qquad S \subseteq \mathcal{D}.$$

For any $S \subseteq T \subseteq \mathcal{D}$ and any $d \in \mathcal{D} \setminus T$ the marginal gain of adding $d$ is

$$\Delta_d(S) \;=\; h_i(S \cup \{d\}) - h_i(S) = \prod_{t \in S}(1 - \mathbf{W}_{t,i}) - \prod_{t \in S}(1 - \mathbf{W}_{t,i})(1 - \mathbf{W}_{d,i}) = \mathbf{W}_{d,i} \prod_{t \in S}(1 - \mathbf{W}_{t,i}).$$

Similarly,

$$\Delta_d(T) = \mathbf{W}_{d,i} \prod_{t \in T}(1 - \mathbf{W}_{t,i}).$$

Since $0 \le 1 - \mathbf{W}_{t,i} \le 1$ for all $t$, we have

$$\prod_{t \in S}(1 - \mathbf{W}_{t,i}) \ge \prod_{t \in T}(1 - \mathbf{W}_{t,i}),$$

and therefore $\Delta_d(S) \ge \Delta_d(T)$. This is exactly the diminishing returns property, so $h_i(\cdot)$ is submodular. Moreover $\Delta_d(S) = \mathbf{W}_{d,i} \prod_{t \in S}(1 - \mathbf{W}_{t,i}) \ge 0$, so $h_i(\cdot)$ is monotone non-decreasing.

**(2) Sum preserves submodularity and monotonicity for Coverage Objective.** The coverage objective $G(S) = \sum_i \mathbf{e}_i h_i(S)$ is a nonnegative linear combination of the functions $h_i$. Nonnegative linear combinations of submodular (resp. monotone) functions remain submodular (resp. monotone) (Iyer, 2015). Hence $G(\cdot)$ is monotone and submodular.

**(3) Per-document term.** Fix a document $d$. Define its disutility score as

$$\mathrm{DisUtil}(d) \;=\; 1 - \max_{e \in \mathcal{E}} \mathbb{P}(d \mid e)\mathbb{P}(e \mid q).$$

Now consider the noise function over a set of documents

$$S \subseteq \mathcal{D} : R(S) \;=\; \sum_{d \in S} \mathrm{DisUtil}(d).$$

Notice that each per-document term $\mathbf{c}_d \cdot \mathrm{DisUtil}(d)$ depends only on $d$ and the evidence set $\mathcal{E}$, but not on the other documents in $S$. Hence, each term contributes additively and independently.

**(4) Submodularity and monotonicity of Noise Objective.** For any $S \subseteq T \subseteq \mathcal{D}$ and any document $d \in \mathcal{D} \setminus T$, the marginal gain of adding $d$ is

$$\Delta_d(S) \;=\; R(S \cup \{d\}) - R(S) \;=\; \mathrm{DisUtil}(d).$$

Similarly,

$$\Delta_d(T) = \mathrm{DisUtil}(d).$$

Since the marginal contribution of $d$ is identical regardless of the context set, the diminishing returns property holds trivially. This means $R(\cdot)$ is modular (a special case of submodular) (Iyer, 2015).

Moreover, as $\mathrm{DisUtil}(d) \ge 0$, we have $\Delta_d(S) \ge 0$, implying monotonicity. Therefore, the noise objective

$$R(S) = \sum_{d \in S} \left( 1 - \max_{e \in \mathcal{E}} \mathbb{P}(d \mid e)\mathbb{P}(e \mid q) \right)$$

is a monotone modular function, and hence also submodular.

**(3) NP-hardness.** Maximizing our proposed objective $g(\cdot)$ under a cardinality constraint is generally NP-hard. A cardinality constraint means that the solution set $S$ is restricted to contain at most $K$ documents (i.e., $|S| \le K$), which reflects the practical setting of re-ranking where we can only select a limited number of documents for the generator.

This complexity includes the well-known Max-k-Cover problem as a special case. Specifically, if each document d either fully covers a particular information requirement $e_i (p_{d,i} = 1)$ or does not cover it at all ($p_{d,i} = 0$), then the coverage function $h_i(S)$ becomes an indicator function that equals 1 if $e_i$ is covered by at least one document in $S$, and 0 otherwise. Under this setting, maximizing $g(S)$ is equivalent to selecting $K$ documents to cover as many weighted information requirements as possible, which is exactly the Max-k-Cover problem, which is known to be NP-hard (Iyer, 2015).

Therefore, even in the more general case where $p_{d,i}$ takes continuous values (representing partial coverage), maximizing $g(\cdot)$ remains NP-hard. This justifies the need for an efficient approximation algorithm, such as our proposed greedy approach. $\qquad\square$

You are given a query and a set of documents.
Your goal is to reason through the documents step by step to answer the query.
**Instructions**:" 1. Only include reasoning steps that clearly contribute to solving the query.
2. Make sure each step is logically connected and leads to the final answer.
3. Use specific evidence from the documents in each step (e.g., facts, names, dates, relationships).
4. In each step, cite document snippets or facts that directly support your reasoning.
5. Avoid vague or redundant steps. Do not repeat the query.
6. Limit reasoning to 5 steps or fewer. Be concise and precise.
**Format**: Each step must start with [Step N]: followed by your reasoning.
Conclude with [Answer]: followed by the final answer (and nothing else).
**Example format**:
[Step 1]: ...
[Step 2]: ...
...
[Answer]: ...

Table 4: Input prompt for goal-oriented reasoning of PureCover

## F  PROMPT DETAILS

## G  LLM USAGE DISCLOSURE

In accordance with ICLR 2026 policy, we disclose our use of large language models (LLMs) in preparing this manuscript. We employed GPT-5 (OpenAI) solely to aid in polishing the writing, specifically for improving clarity, grammar, and sentence structure across sections. All technical content, algorithmic contributions, experimental results, and scientific conclusions remain entirely the authors' own work, without any LLM involvement.

