# OpenReview forum: "PureCover: Bridging the Gap in Re-ranking for Retrieval-Augmented Generation via Balancing Coverage and Noise"
_ICLR.cc/2026/Conference — ICLR 2026 Conference Withdrawn Submission_

### Official Review · Reviewer_waZd · 2025-10-26

**Soundness:** 2
**Presentation:** 3
**Contribution:** 2
**Rating:** 4
**Confidence:** 2

**Summary:**

PureCover reframes RAG re-ranking as selecting a set that jointly maximizes evidence coverage while minimizing noise, rather than ranking by per-doc relevance or diversity; the system estimates per-query information needs from CoT reasoning attention, calibrates away position bias, solves a submodular 0–1 objective with a greedy procedure, then distills the selection into a lightweight LLM selector that filters by first-token logits at inference.

**Strengths:**

1. LLMs lack human selective attention, so RAG needs a list-level objective balancing coverage and noise rather than pure relevance. The paper formalizes this and ties it to CoT-based estimation.

2. PureCover edges strong cross-encoder and LLM rankers on all four datasets.

**Weaknesses:**

1. end-to-end tables fix K=5 for all datasets, but several baselines are K-sensitive; the paper only shows K-sweep on HotpotQA. Best-K comparisons and Pareto fronts (EM/F1 vs #docs) across datasets would be more convincing.


2. Information Coverage/Purity partly rely on “contains answer string” heuristics; multi-hop bridging evidence may be under-counted. Please validate these proxies with human-labeled supports on a sample.

3. experiments use one retriever (e5-base-v2) and one generator setup; results may depend on the retrieval pool and generator size. Cross-retriever/generator tests would strengthen claims.

**Questions:**

see weakness

---

> ### Author Response · Authors · 2025-11-21
> **Response to Reviewer waZd**
>
> Thank you very much for your valuable comments and suggestions. They are extremely helpful and have greatly contributed to improving the quality of our work.
>
> > Q1: The paper only shows K-sweep on HotpotQA. Best-K comparisons and Pareto fronts (EM/F1 vs #docs) across datasets would be more convincing.
>
> Thank you for the suggestion. Due to space limitations, we only report the K-sweep results on HotpotQA in the main text. The K-sweep results for the other datasets are provided in the appendix; please refer to Appendix C.2 (Line 939) for more experimental results.
>
> Following your suggestion, we conduct a fair comparison and reported the Best-K performance of different methods, presented through Pareto fronts (EM/F1 vs. #docs) across datasets. You can refer to Line 447 in the main text and the table below.
>
> | dataset  | model           | metric | #Docs | EM     | F1     |
> |----------|-----------------|--------|-------|--------|--------|
> | HotpotQA | PureCover(ours) | EM     | 3.45  | 0.3800 | 0.4900 |
> | HotpotQA | xQuAD           | EM     | 5     | 0.3600 | 0.4710 |
> | HotpotQA | First           | EM     | 5     | 0.3580 | 0.4740 |
> | HotpotQA | ICR             | EM     | 5     | 0.3540 | 0.4659 |
> | HotpotQA | RankZephyer     | EM     | 10    | 0.3660 | 0.4680 |
> | HotpotQA | SETR            | EM     | 2.65  | 0.3660 | 0.4562 |
>
>
> | dataset | model           | metric | #Docs | EM     | F1     |
> |---------|-----------------|--------|-------|--------|--------|
> | 2Wiki   | PureCover(ours) | EM     | 3.04  | 0.3340 | 0.4080 |
> | 2Wiki   | xQuAD           | EM     | 5     | 0.3180 | 0.3955 |
> | 2Wiki   | First           | EM     | 15    | 0.3120 | 0.3892 |
> | 2Wiki   | ICR             | EM     | 10    | 0.3000 | 0.3798 |
> | 2Wiki   | RankZephyer     | EM     | 15    | 0.3120 | 0.3896 |
> | 2Wiki   | SETR            | EM     | 3.42  | 0.3240 | 0.3894 |
>
> | dataset | model           | metric | #Docs | EM     | F1     |
> |---------|-----------------|--------|-------|--------|--------|
> | musique | PureCover(ours) | EM     | 2.54  | 0.1480 | 0.2344 |
> | musique | xQuAD           | EM     | 5     | 0.1260 | 0.2052 |
> | musique | First           | EM     | 15    | 0.1320 | 0.2236 |
> | musique | ICR             | EM     | 15    | 0.1400 | 0.2291 |
> | musique | RankZephyer     | EM     | 3     | 0.1440 | 0.2387 |
> | musique | SETR            | F1     | 2.03  | 0.1460 | 0.2265 |
>
>
> | dataset    | model           | metric | #Docs | EM     | Acc    |
> |------------|-----------------|--------|-------|--------|--------|
> | strategyqa | PureCover(ours) | EM     | 2.34  | 0.7080 | 0.7240 |
> | strategyqa | xQuAD           | EM     | 15    | 0.7100 | 0.7240 |
> | strategyqa | First           | EM     | 15    | 0.7040 | 0.7160 |
> | strategyqa | ICR             | EM     | 15    | 0.7180 | 0.7220 |
> | strategyqa | RankZephyer     | EM     | 15    | 0.7060 | 0.7140 |
> | strategyqa | SETR            | EM     | 3.14  | 0.7060 | 0.7200 |
>
> The results show that our method achieves better Pareto efficiency than the baselines on almost all datasets, attaining higher QA performance while reducing generator token usage.

---

> ### Author Response · Authors · 2025-11-21
> **Response2  to Reviewer waZd**
>
> > Q2: Information Coverage/Purity partly rely on “contains answer string” heuristics. Please validate these proxies with human-labeled supports on a sample.
>
> Thank you for your valuable suggestion. We want to address your concern from two perspectives.
>
> **Existing studies have demonstrated the validity of this evaluation.** First, because the documents retrieved in the RAG process often lack golden labels (e.g., annotations indicating whether a document is related or unrelated), using the “contains answer string” heuristic has been widely adopted as a reasonable proxy for estimating whether a document contains important evidence. This approach has been highlighted and validated in previous RAG studies [1,2], supporting its effectiveness for evaluating document relevance in the absence of explicit annotations.
>
>
> **Experiment on human-annotated supports**.
>
> Following your suggestion, we conducted experiments on the HotpotQA dataset using the coverage-aware reranking baseline that performed best in our method. We then recalculated Recall, Precision, and Hit Rate (HR) using human-annotated evidence.
> Due to time constraints for the rebuttal, we will continue to complete experiments on all baselines in the future and include the results in an updated version of the paper.
>
> | Model | EM     | F1          | Recall | Precision | HR     |
> |--------|--------|--|--|----|--------|
> | PureCover        | 0.3800 | 0.4864      | 0.3531 | 0.2880    | 0.8655 |
> | MMR                    | 0.3560 | 0.4734      | 0.3482 | 0.2590    | 0.8418 |
> | xQuAD                  | 0.3700 | 0.4801      | 0.3453 | 0.2619    | 0.8320 |
> | SETR | 0.3460 | 0.4562      | 0.3180 | 0.2512    | 0.8198 |
>
> As shown in the table, our method consistently outperforms the baselines in both end-to-end RAG evaluation and ranking evaluation based on human-annotated evidence.
>
>
>
> [1] Jiajie Jin, Yutao Zhu, Xinyu Yang, Chenghao Zhang, and Zhicheng Dou. Flashrag: A modular
> toolkit for efficient retrieval-augmented generation research. arXiv preprint arXiv:2405.13576,
> 2024
>
> [2] Xanh Ho, et al. Constructing a multi-
> hop QA dataset for comprehensive evaluation of reasoning steps. ICCL 2020,

---

> ### Author Response · Authors · 2025-11-21
> **Response3 to Reviewer waZd**
>
> > Q3: Experiments use one retriever (e5-base-v2) and one generator setup; results may depend on the retrieval pool and generator size. Cross-retriever/generator tests would strengthen claims.
>
> **Experimental result across retriever.** Thank you for the suggestion. **We have already conducted experiments with different widely-used retriever (the sparse retriever BM25) in the paper**, but due to space limitations, we placed the results in the appendix (see Appendix C and Table 3). For your convenience, we provide it below for your reference.
>
>
> |      ~    | HotpotQA |        | 2Wiki  |        | Musique |        | StrategyQA |        |
> |------|---|------|--------|----|---------|--------|------------|--------|
> | Model    | EM  | F1     | EM     | F1     | EM      | F1     | EM         | Acc    |
> | Retrieval Only     | 30.80     | 40.63  | 28.40  | 34.08  | 7.40    | 16.12  | 62.20      | 65.40  |
> | Bge-Reranker-Large | 36.60   | 47.22  | 32.60  | 39.52  | 10.20   | 18.93  | 65.00      | 66.00  |
> | RADIO              | 34.60  | 46.01  | 28.80  | 36.96  | 10.20   | 20.39  | 67.00      | 69.60  |
> | Qwen3-Reranker-8B  | 34.40  | 46.19  | 29.60  | 36.18  | 9.80    | 18.75  | 65.40      | 67.20  |
> | RankVicuna   | 34.00  | 44.69  | 27.40  | 33.82  | 9.40    | 18.27  | 64.60      | 66.60  |
> | RankLlama          | 34.40    | 46.06  | 28.40  | 34.97  | 9.80    | 18.74  | 64.60      | 66.80  |
> | RankZephyr         | 33.20    | 44.93  | 30.80  | 37.28  | 11.20   | 19.77  | 64.00      | 65.80  |
> | FIRST              | 33.40         | 45.08  | 30.40  | 37.26  | 11.00   | 19.34  | 64.20      | 67.40  |
> | ICR                | 35.20          | 46.22  | 31.00  | 37.45  | 8.80    | 17.15  | 62.40      | 65.00  |
> | RankGPT            | 35.00        | 46.49  | 33.20  | 39.89  | 10.80   | 19.04  | 64.60      | 66.80  |
> | IA-Select          | 34.00  | 43.66  | 28.20  | 35.62  | 8.60    | 17.92  | 60.40      | 63.80  |
> | MMR                | 34.60   | 45.75  | 31.00  | 37.52  | 11.00   | 19.69  | 62.20      | 64.20  |
> | xQuAD              | 34.20     | 45.07  | 28.60  | 34.96  | 8.40    | 17.45  | 62.20      | 64.40  |
> | SETR               | 36.80     | 47.36  | 31.20  | 38.36  | 13.80   | 22.65  | 70.00      | 71.80  |
> | PureCover(ours)    | **37.80 **  | **47.74**  | **34.00**  | **40.30**  | **15.20**   | **23.39**  | $\underline{69.60}$     |  $\underline{68.00}$  |
>
> **Experimental result across generator.** According to your valuable suggestion, **we also conducted experiments using different generators (LLaMa3-8B) to verify the generalization and effectiveness of our method**. Due to time constraints during the discussion period, we reproduced the most important baselines. We will continue to complete experiments on all baselines in the future and include the results in an updated version of the paper.
>
>
> | Model     | HotpotQA |        | 2WikiMultiHopQA |        | MusiQue |        | StrategyQA |        |
> |------|--|--------|---|----|---|--------|----|--------|
> |                                | EM       | F1     | EM              | F1     | EM      | F1     | EM         | Acc    |
> | bm25                           | 25.60    | 35.41  | 11.60           | 21.19  | 6.00    | 13.36  | 47.60      | 63.60  |
> | e5-base-v2                     | 27.00    | 37.49  | 12.00           | 23.00  | 8.60    | 15.96  | 58.00      | 67.40  |
> | Bge-Reranker-Large             | 31.00    | 42.26  | 13.00           | 23.39  | 10.60   | 18.97  | 58.40      | 66.00  |
> | Qwen3-Reranker                 | 30.00    | 41.27  | 14.20           | 24.61  | 10.80   | 19.83  | 60.20      | 67.60  |
> | RankLlama                      | 30.40    | 41.12  | 14.40           | 24.90  | 10.20   | 17.51  | 58.00      | 66.80  |
> | RankZephyr      | 28.60    | 39.55  | 12.40    | 23.74  | 10.20   | 18.90  | 55.60      | 65.40  |
> | FIRST                          | 30.40    | 41.44  | 13.40           | 23.22  | 10.20   | 18.43  | 58.00      | 66.60  |
> | ICR                            | 30.00    | 40.41  | 12.80           | 22.52  | 9.80    | 17.32  | 60.40      | 68.40  |
> | MMR                            | 29.20    | 39.90  | 9.80 | 21.26  | 9.20    | 16.64  | 60.00      | 66.20  |
> | xQuAD                          | 24.40    | 34.03  | 9.60    | 20.22  | 7.60    | 15.00  | 54.80      | 64.20  |
> | SETR                           | 30.20    | 40.03  | 13.20           | 22.42  | 8.80    | 16.52  | 61.00      | 67.80  |
> | PureCover | **31.40**    | **41.76**  | **15.40**           |**25.01**  | **11.20**   | 18.41  | **63.40**      | **69.00**  |
>
>
> As shown by the experimental results in the table, **our method achieves RAG performance that surpasses the baselines across different retrievers and generators, demonstrating its robustness and generalizability**. These results indicate that our approach is effective regardless of the specific choice of retrieval or generation components, highlighting the adaptability of our attention-based reranking mechanism in diverse RAG settings.

---

### Official Review · Reviewer_rmWK · 2025-10-29

**Soundness:** 2
**Presentation:** 2
**Contribution:** 2
**Rating:** 4
**Confidence:** 2

**Summary:**

This paper introduces PureCover, a novel document re-ranking framework for Retrieval-Augmented Generation (RAG) that reframes document selection as a multi-objective optimization problem to balance information coverage and noise minimization. Unlike traditional re-rankers designed for human users, PureCover leverages LLMs’ internal attention patterns during goal-oriented reasoning to identify key evidence and employs a greedy algorithm with set-wise distillation for efficient inference. Experiments on multiple multi-hop QA benchmarks show that PureCover consistently outperforms state-of-the-art baselines by better balancing coverage and noise, leading to significant improvements in RAG performance.

**Strengths:**

1. Clear Problem Formulation: It identifies a specific, often-overlooked flaw in applying traditional re-rankers to RAG systems.

2. Strong Empirical Results: It demonstrates consistent and significant performance improvements over many state-of-the-art baselines.

**Weaknesses:**

Limited Justification for Core Assumptions: The entire method hinges on using LLM attention during Chain-of-Thought (CoT) reasoning as a proxy for "information requirements" and document utility. While the results are positive, the paper provides no validation that these attention patterns are a reliable and generalizable ground truth for what constitutes essential evidence versus noise. This is a significant assumption. A stronger justification, such as a human annotation study correlating high-attention tokens with key facts, or an ablation showing that the content of the CoT steps (rather than just the attention signal) is crucial, would make the foundational premise more convincing.

**Questions:**

See weaknesses.

---

> ### Author Response · Authors · 2025-11-21
> **Response to Reviewer rmWK**
>
> Thank you very much for your valuable comments and suggestions. They are extremely helpful and have greatly contributed to improving the quality of our work.
>
> > Q1: Lack of validation that these attention patterns are a reliable and generalizable ground truth for what constitutes essential evidence versus noise.
>
> Thank you for your valuable suggestion. We will do our best to address your concerns in the following.
>
> **Reliability of attention as re-ranking signal**. We apologize for not explicitly discussing in the paper that attention patterns have been extensively explored and validated for their reliability in previous research. Prior research has shown that attention patterns in LLMs provide a highly informative signal for identifying and extracting key information from the context [1,2,3,4], which can be effectively leveraged for re-ranking. For example, as shown in the experiments of [1], calibrated attention patterns can achieve zero-shot ranking performance that even surpasses methods relying on powerful LLMs for direct ranking (e.g., RankGPT using GPT-3.5-turbo). This indicates that attention patterns constitute a strong and valuable signal for ranking.
> Similarly, other studies [3] have empirically shown that calibrated attention weights can effectively identify the most relevant documents in the context. Leveraging this property, several works have used attention to enable document pruning in RAG systems.
>
> Building on the prior work’s validation of the reliability and generalizability of attention weights, our method uses attention signals to:
> (1) identify the importance of different information needs within the query, and
> (2) assess how each document contributes to these information needs, thereby supporting a more effective coverage-aware reranking process.
>
> **Evidence beyond downstream RAG performance.**
> Beyond downstream RAG metrics, we also evaluate our method using several retrieval-level intermediate metrics to demonstrate that attention signals can effectively filter noisy documents and select key evidence.
>
> As shown in Fig. 3(c–d) (Line 387) in the main text, the documents selected using our attention-based method consistently outperform baselines across different values of K in terms of both Recall (information coverage) and Precision (information purity).
>
> These results further confirm that attention signals provide a reliable mechanism for extracting essential information, independent of downstream generator behavior.
>
>
>
> [1] Shijie Chen, et al. Attention in large language models yields efficient zero-shot re-rankers. ICLR 2025
>
> [2] Peysakhovich, et al. Attention sorting combats recency bias in long context language models. arXiv preprint arXiv:2310.01427
>
> [3] Hsieh, Cheng-Yu, et al. Found in the middle: Calibrating positional attention bias improves long context utilization. ACL 2024.
>
> [4] Fang, Yixiong, et al. Attentionrag: Attention-guided context pruning in retrieval-augmented generation. arXiv preprint arXiv:2503.10720 (2025).

---

> ### Author Response · Authors · 2025-11-21
> **Response2 to Reviewer rmWK**
>
> > Q1: Lack of validation that these attention patterns are a reliable and generalizable ground truth for what constitutes essential evidence versus noise.
>
> **Additional experiments on reliability and generalizability.**
> Following your valuable suggestion, we conduct further experiments to evaluate the reliability and generalizability of attention patterns in LLMs for re-ranking. These experiments examine whether attention-based signals remain stable and effective across different CoT prompts, temperatures(0-1), and model scales (30B, 7B, 3B).
>
> *Experimental setups.* Since the information requirements $e$ may vary across different query, as well as CoT prompts, temperature settings, and model scales, directly comparing P(e| q) and P(d | e) among these setting is challenging. Hence, we instead evaluate the resulting P(d | q) produced by the optimization solver with the same hyper-parameter $\lambda$ maintiained. Since the optimization solver takes P(e| q) and P(d | e) as inputs and outputs P(d | q), we can assess the stability of the LLM-attention-derived P(e | q) and P(d | e) by examining the resulting behavior of P(d|q).
>
> Following you advice, Recall, Precision, and Hit Rate are computed using human-annotated key-evidence documents. These results confirm that attention patterns remain stable and effective across diverse prompting and generation configurations.
>
> 1. Different CoT prompt
>
> | Prompt  | EM     | F1     | Recall | Precision | HR     |
> |--|-|--|--|---|----|
> | Prompt1 | 0.3800 | 0.4864 | 0.3531 | 0.2880    | 0.8655 |
> | Prompt2 | 0.3720 | 0.4825 | 0.3220 | 0.2689    | 0.8289 |
> | Prompt3 | 0.3680 | 0.4782 | 0.3419 | 0.2743 | 0.8582 |
> | MMR     | 0.3560 | 0.4734 | 0.3482 | 0.2590| 0.8418 |
> | xquad   | 0.3700 | 0.4801 | 0.3453 | 0.2619 | 0.8320 |
> | SETR    | 0.3460 | 0.4562 | 0.3180 | 0.2512    | 0.8198 |
>
> As shown in the table, our model exhibits stable RAG performance across different prompts. Even under the simplest prompt configuration (Given a query and a set of documents, reason step-by-step using only logically necessary and evidence-supported steps to produce the final answer), the model is still able to achieve strong RAG performance.
>
> Also, we find that higher ranking metrics do not always lead to better end-to-end generation quality. Some documents, though containing key information, may be less LLM-friendly and hinder generation, while others, even if not directly relevant, help connect information. By leveraging attention signals, our method effectively identifies these LLM-friendly documents, supporting improved comprehension and higher-quality generation.
>
> 2. Temperatrue Values
>
> | Temperature | EM | F1 | Recall | Precision | HR     |
> |--|--|--|--|----|--|
> | 0  | 0.3800 | 0.4864 | 0.3531 | 0.2880    | 0.8655 |
> | 0.2  | 0.3760 | 0.4830 | 0.3412 | 0.2733    | 0.8631 |
> | 0.4  | 0.3720 | 0.4889 | 0.3463 | 0.2807    | 0.8631 |
> | 0.6  | 0.3680 | 0.4809 | 0.3400 | 0.2807    | 0.8631 |
> | 0.8 | 0.3780 | 0.4848 | 0.3469 | 0.2768    | 0.8606 |
> | 1 | 0.3600 | 0.4713 | 0.3314 | 0.2675    | 0.8533 |
> | MMR| 0.3560 | 0.4734 | 0.3482 | 0.2590    | 0.8418 |
> | xQuAD| 0.3700 | 0.4801 | 0.3453 | 0.2619    | 0.8320 |
> | SETR   | 0.3460 | 0.4562 | 0.3180 | 0.2512    | 0.8198 |
>
> Across different temperature settings, we observe that our method maintains strong performance at several temperatures and surpasses traditional coverage-aware baselines under most temperature configurations.
>
> 3. Model scales
> | Model-size  | EM | F1  | Recall | Precision | HR     |
> |---|--|--|---|----|--|
> | Qwen2.5-32B  | 0.3800 | 0.4864      | 0.3531 | 0.2880    | 0.8655 |
> | Qwen2.5-7B| 0.3680 | 0.4772      | 0.3480 | 0.2712    | 0.8598 |
> | Qwen2.5-3B | 0.3720 | 0.4889      | 0.3430 | 0.2787    | 0.8509 |
> | MMR     | 0.3560 | 0.4734  | 0.3482 | 0.2590    | 0.8418 |
> | xQuAD  | 0.3700 | 0.4801      | 0.3453 | 0.2619    | 0.8320 |
> | SETR(GPT-4o distilled) | 0.3460 | 0.4562      | 0.3180 | 0.2512    | 0.8198 |
>
> We conduct experiments across different model sizes and were surprised to find that using attention as the signal for re-ranking does not always rely on larger models. In fact, we observed that the 3B model can sometimes achieve even better RAG performance than the 7B model, indicating that our method generalizes well across models of different sizes.
>
> In summary, the attention signal not only **achieves comparable or better ranking performance than traditional embedding-based baselines** (e.g., MMR, XQuAD)  or directly on the LLM’s selection mechanism (e.g., SETR), but also **exhibits stronger reliability and generalizability (prompt, temperature, mode size) for RAG**. Unlike conventional relevance-based reranking, it can selects LLM-friendly documents, enabling higher-quality generation.

---

### Official Review · Reviewer_xVuM · 2025-11-02

**Soundness:** 3
**Presentation:** 3
**Contribution:** 2
**Rating:** 4
**Confidence:** 3

**Summary:**

The paper argues IR re-rankers assume human selective attention while LLMs don’t, so RAG needs re-ranking that balances evidence coverage with noise suppression.
It defines coverage and noise for complex QA, then reframes document selection as a multi-objective problem solved with calibrated reasoning-attention signals.

**Strengths:**

The goal is RAG-native and practical: cover what’s needed and keep junk out of the context.

The position-bias calibration acknowledges “lost-in-the-middle” issues and corrects attention before using it.

The set-wise distillation and first-token scoring make deployment feel realistic instead of research-only.

**Weaknesses:**

The method leans hard on CoT prompting and attention introspection, which can drift across models and prompts.

The noise term uses a max over requirements, which might miss partial usefulness or cross-requirement interactions.

Submodularity and monotonicity hinge on attention-based estimates that could be noisy in practice.

The calibration relies on dummy CoTs/docs and may not transfer cleanly to other layout or packing strategies.

The distilled selector introduces thresholds and budgets that still need tuning in new domains.

**Questions:**

How stable are P(e|q) and P(d|e) across different CoT prompts, temperatures, and model sizes.

What’s a practical recipe to set lambda and the selection threshold without dataset-specific sweeps.

Can the framework capture “two-docs-together” interactions rather than “at least one doc” coverage only.

---

> ### Author Response · Authors · 2025-11-21
> **Response to Reviewer xVuM**
>
> Thank you very much for your valuable comments and suggestions. They are extremely helpful and have greatly contributed to improving the quality of our work.
>
>
> > Q1: How stable are P(e|q) and P(d|e) across different CoT prompts, temperatures, and model sizes.
>
> We appreciate your insightful question and apologize for not making this aspect sufficiently clear in the paper. Below, we provide experimental evidence to address your question.
>
> **Directly evaluating P(e|q) and P(d|e) is difficult**. Because the information requirements $e$ may vary across different queries, as well as CoT prompts, temperature settings, and model scales, directly comparing P(e| q) and P(d | e) among these settings is challenging. Hence, we instead evaluate the resulting P(d | q) produced by the optimization solver with the same hyper-parameter $\lambda$ maintained. Since the optimization solver takes P(e| q) and P(d | e) as inputs and outputs P(d | q), we can assess the stability of the LLM-attention-derived P(e | q) and P(d | e) by examining the resulting behavior of P(d| q).
>
> **Experimental results.** As shown in the tables below, we conduct additional analyses to examine the robustness of the document-ranking probabilities P(d | q) under (1) different CoT prompt variants, (2) temperature values (0.0–1.0), and (3) model scales (3B, 7B, and 30B). We use human-annotated key evidence documents as positive examples, and measure Recall, Precision, and Hit Rate (HR) to evaluate whether PureCover can consistently and correctly identify these relevant documents. In addition, we also report downstream RAG performance using EM and F1. Due to time constraints, we conducted our experiments on HotpotQA. In the future, we will extend our evaluation to additional datasets and update them in the revised version of paper.
>
> 1. CoT prompt
>
> | Prompt  | EM     | F1     | Recall | Precision | HR     |
> |---|----|--|---|-|--|
> | Prompt1 (ours) | 0.3800 | 0.4864 | 0.3531 | 0.2880    | 0.8655 |
> | Prompt2 (simple) | 0.3720 | 0.4825 | 0.3220 | 0.2689    | 0.8289 |
> | Prompt3 (very simple) | 0.3680 | 0.4782 | 0.3419 | 0.2743    | 0.8582 |
> | MMR     | 0.3560 | 0.4734 | 0.3482 | 0.2590    | 0.8418 |
> | xQuAD   | 0.3700 | 0.4801 | 0.3453 | 0.2619    | 0.8320 |
> | SETR    | 0.3460 | 0.4562 | 0.3180 | 0.2512    | 0.8198 |
>
>
> As shown in the table, our model exhibits stable and competitive RAG performance across different prompts. Even under the simplest prompt configuration (*Given a query and a set of documents, reason step-by-step using only logically necessary and evidence-supported steps to produce the final answer.*), the model maintains strong performance, demonstrating its robustness and effectiveness across diverse prompting conditions.
>
>
> 2. Temperatrue Values
>
> | Temperature | EM     | F1     | Recall | Precision | HR     |
> |--|--|--|--|-|-|
> | 0           | 0.3800 | 0.4864 | 0.3531 | 0.2880    | 0.8655 |
> | 0.2         | 0.3760 | 0.4830 | 0.3412 | 0.2733    | 0.8631 |
> | 0.4         | 0.3720 | 0.4889 | 0.3463 | 0.2807    | 0.8631 |
> | 0.6         | 0.3680 | 0.4809 | 0.3400 | 0.2807    | 0.8631 |
> | 0.8         | 0.3780 | 0.4848 | 0.3469 | 0.2768    | 0.8606 |
> | 1           | 0.3600 | 0.4713 | 0.3314 | 0.2675    | 0.8533 |
> | MMR         | 0.3560 | 0.4734 | 0.3482 | 0.2590    | 0.8418 |
> | xQuAD       | 0.3700 | 0.4801 | 0.3453 | 0.2619    | 0.8320 |
> | SETR        | 0.3460 | 0.4562 | 0.3180 | 0.2512    | 0.8198 |
>
> Across different temperature settings, we observe that our method maintains strong performance at several temperatures and surpasses traditional coverage-aware baselines under most temperature configurations.
>
>
> 3. Model scales
> | Model size | EM     | F1          | Recall | Precision | HR     |
> |--------|--------|--|--|----|--------|
> | Qwen2.5-32B            | 0.3800 | 0.4864      | 0.3531 | 0.2880    | 0.8655 |
> | Qwen2.5-7B             | 0.3680 | 0.4772      | 0.3480 | 0.2712    | 0.8598 |
> | Qwen2.5-3B             | 0.3720 | 0.4889      | 0.3430 | 0.2787    | 0.8509 |
> | MMR                    | 0.3560 | 0.4734      | 0.3482 | 0.2590    | 0.8418 |
> | xQuAd                  | 0.3700 | 0.4801      | 0.3453 | 0.2619    | 0.8320 |
> | SETR | 0.3460 | 0.4562      | 0.3180 | 0.2512    | 0.8198 |
>
> Across prompts, temperature settings, and model scales, our method demonstrates strong robustness and generalization. The RAG performance is competitive with some baselines even under the simplest prompting conditions, and our method generally outperforms traditional coverage-aware baselines across a majority of temperature configurations. **Moreover, our experiments show that attention-based re-ranking is not largely dependent on larger models**: even smaller models (e.g., 3B) can perform on par with or even better than larger ones (7B), highlighting the efficiency and effectiveness of attention as a re-ranking signal.   We hope these findings can offer new insights to the RAG community.

---

> ### Author Response · Authors · 2025-11-21
> **Response 2 to Reviewer xVuM**
>
> > Q2: What’s a practical recipe to set lambda and the selection threshold without dataset-specific sweeps.
>
> Thank you for your valuable suggestions, which have been very helpful for our work.
>
>
>  **Trade-off coefficient $\lambda$**
> 1. First, we would like to emphasize that our method is highly robust to the hyperparameter $\lambda$. As shown in Figure 4(a) of the paper, compared with traditional coverage-aware baselines, which exhibit large fluctuations as $\lambda$ varies, our method demonstrates consistently strong performance across different $\lambda$ values. This indicates that our approach is not overly sensitive to the choice of $\lambda$ and can reliably maintain high performance under different values.
>
> 2. During our experiments, we observed that the optimal choice of $\lambda$ does not vary significantly across different multi-hop QA datasets. This is illustrated in the experimental results shown in the figure below.
>
> |   | HotpotQA | 2Wiki | Musique |StrategyQA |
> |---|----------|:------|:--------|:-----|
> | best_lambda | 0.3      | 0.2   | 0.2     | 0.3|
>
> Therefore, we do not need to specifically tune $\lambda$ for different multi-hop QA datasets; the same $\lambda$ can generalize well across multiple datasets. We encourage future work to adopt this $\lambda$ setting as well.
>
> **Selection threshold $\tau_3$.** We apologize for not explicitly highlighting the selection threshold $\tau_3$ in the paper. Regarding this threshold, we follow prior work [1] and set $\tau_3 = 0.5$ across different datasets. We found that this value yields the best performance and remains effective across datasets, without the need for dataset-specific adjustments. We encourage future work to adopt this threshold setting as well.
>
> [1] Qu, Changle, et al. Uplift-RAG: Uplift-Driven Knowledge Preference Alignment for Retrieval-Augmented Generation.EMNLP 2025.
>
>
>
>
> > Q3: Can the framework capture “two-docs-together” interactions rather than “at least one doc” coverage only.
>
> Thank you for your valuable suggestion; this is indeed an important point.
>
> When modeling information coverage, we decompose a complex user query into several finer-grained atomic information needs, denoted as $e$. To enable the LLM generator to fully answer the query, the set of documents in the reranked list must collectively cover these information needs. If any information need is not satisfied, the model cannot generate a complete or correct answer.
>
> For example, for a query containing information needs A and B, we aim to ensure that at least one document in the reranked list satisfies A and at least one satisfies B. When all information needs are covered by at least one document, these documents collectively create a synergistic effect at the information level, jointly supporting the LLM’s answer.
>
> In other words, although our framework does not explicitly model a “two-docs-together” interaction term, it implicitly captures the complementarity and collaborative relationships between documents through joint coverage at the level of information needs.

---

### Note · Authors · 2026-01-23

**Comment:**

Dear Program Chairs, Area Chair, and Reviewers,

We would like to formally request permission to withdraw our submission to ICLR 2026 entitled
“PureCover: Bridging the Gap in Re-ranking for Retrieval-Augmented Generation via Balancing Coverage and Noise”
(Submission Number: 16803).

The full paper submission deadline for SIGIR 2026 is January 22. However, due to the delayed notification timeline of ICLR 2026, the decision for this submission will not be available before that deadline. As a result, it is not feasible for us to wait for the ICLR decision while meeting the SIGIR submission deadline. Therefore, we kindly ask for your understanding and permission to withdraw this submission in order to proceed with submission to SIGIR 2026.

This request is solely due to scheduling constraints and is independent of the review outcome. We sincerely appreciate the time and effort invested by the reviewers and the area chair in evaluating our work.

We apologize for any inconvenience this withdrawal request may cause and thank you very much for your consideration.

Sincerely,
On behalf of all authors

**Withdrawal Confirmation:**

I have read and agree with the venue's withdrawal policy on behalf of myself and my co-authors.